# Enhancing Men's Awareness of Testicular Diseases (E-MAT) using virtual reality: A randomised pilot feasibility study and mixed method process evaluation

**Mohamad M. Saab**[1☯]*, **Megan McCarthy**[1☯], **Martin P. Davoren**[2,3], **Frances Shiely**[3,4], **Janas M. Harrington**[3,5], **Gillian W. Shorter**[6], **David Murphy**[7], **Billy O'Mahony**[1,7], **Eoghan Cooke**[1,4,8], **Aileen Murphy**[9], **Ann Kirby**[9], **Michael J. Rovito**[10], **Steve Robertson**[11], **Serena FitzGerald**[1], **Alan O'Connor**[12], **Mícheál O'Riordan**[13], **Josephine Hegarty**[1‡], **Darren Dahly**[3,4‡]

1 Catherine McAuley School of Nursing and Midwifery, University College Cork, Cork, Ireland, 2 Sexual Health Centre, Cork, Ireland, 3 School of Public Health, University College Cork, Cork, Ireland, 4 Health Research Board Clinical Research Facility, University College Cork, Cork, Ireland, 5 Centre for Health and Diet Research, School of Public Health, University College Cork, Cork, Ireland, 6 Drug and Alcohol Research Network, School of Psychology, Queen's University Belfast, Belfast, Northern Ireland, United Kingdom, 7 School of Computer Science and Information Technology, University College Cork, Cork, Ireland, 8 Health Research Board National Clinical Trials Office, College of Medicine and Health, University College Cork, Cork, Ireland, 9 Department of Economics, Cork University Business School, University College Cork, Cork, Ireland, 10 Department of Health Sciences, College of Health Professions and Sciences, University of Central Florida, Orlando, Florida, United States of America, 11 School of Allied Health Professions, Nursing & Midwifery, Faculty of Health, University of Sheffield, Sheffield, United Kingdom, 12 St. Finbarr's National Hurling & Football Club, Cork, Ireland, 13 Grenagh GAA Club, Cork, Ireland

☯ These authors contributed equally to this work.
‡ JH and DD authors also contributed equally to this work.
* msaab@ucc.ie

## Abstract

### Introduction

Testicular cancer is among the most common malignancies in men under the age of 50 years. Most testicular symptoms are linked to benign diseases. Men's awareness of testicular diseases and testicular self-examination behaviours are suboptimal. In this pilot feasibility study and process evaluation we examine the feasibility of conducting a future definitive randomised controlled trial (RCT) to test the effect of the Enhancing Men's Awareness of Testicular Diseases using Virtual Reality intervention (E-MAT$_{VR}$) compared to the Enhancing Men's Awareness of Testicular Diseases using Electric information control (E-MAT$_E$). The study protocol is registered on ClinicalTrials.gov (NCT05146466).

### Methods

Male athletes, engaged in Gaelic games, and aged 18 to 50 years were included. Recruitment was via Facebook™, X™ (formerly Twitter™), and posters. Participants were individually randomised to either E-MAT$_{VR}$ or E-MAT$_E$. Data were collected at baseline (T0),

**Data Availability Statement:** The data that support the findings of this study are openly available in the Open Science Foundation at https://osf.io/m5wb7/.

**Funding:** Health Research Board Definitive Interventions and Feasibility Awards (DIFA-2020-028). The funders had no role in study design, data collection and analysis, decision to publish, or preparation of the manuscript.

**Competing interests:** The authors have declared that no competing interests exist.

immediately post-test (T1), and three months post-test (T2) using surveys. Qualitative interviews were conducted with participants and researchers.

## Results

Data were collected from 74 participants. Of those, 66 were retained. All E-MAT$_{VR}$ participants and most E-MAT$_E$ participants (n = 33, 89.2%) agreed/strongly agreed that the device was easy to use and that they were engaged to learn by the device. Most E-MAT$_{VR}$ participants (n = 34, 91.9%) and all E-MAT$_E$ participants agreed/strongly agreed that the time it took them to complete the intervention was reasonable. All 74 participants were extremely satisfied/somewhat satisfied with their overall participation in the study. E-MAT$_{VR}$ was described as interactive, easy, fun, and close to real life. Initial difficulty using VR equipment, nausea, and technical issues were identified as challenges to engaging with E-MAT$_{VR}$. Recommendations were made to make VR more accessible, shorten the survey, and incorporate more interactivity. Across all participants, mean testicular knowledge scores (range 0–1) increased from 0.4 (SD 0.2) at T0 to 0.8 (SD 0.2) at T1. At T2, overall mean scores for participants were 0.7 (SD 0.2). Mean knowledge scores did not differ by trial arm at any timepoint. At T2, all E-MAT$_{VR}$ participants and 29/32 E-MAT$_E$ participants (90.6%) reported purposefully examining their testes within the past three months.

## Conclusion

Findings are promising, highlighting the feasibility of using VR to promote young athletes' awareness of testicular diseases. Considering the strengths, limitations, and lessons learned from this study, some modifications are required prior to conducing an RCT. These include but are not limited to shortening survey questions, incorporating more interactivity and visual content, and targeting more heterogenous male-dominated environments.

## Introduction

Testicular cancer is one of the most common malignancies in men under 50 years of age [1]. The incidence of testicular cancer has doubled over the last four decades [2]. In Ireland, where our study was conducted, 176 men develop testicular cancer (2.4% increase per year) and 7 men die from it each year, with 91% of cases and 75% of deaths occurring in men younger than 50 years [3].

Histopathology reports of 215 orchidectomies (i.e., surgical removal of the affected testis) performed between 1975–1985 and 2007–2012 found that tumour size was reduced significantly [4]. This was attributed to raising testicular cancer awareness in the United Kingdom (UK) [4]. While orchidectomy remains the primary curative treatment for testicular cancer, other treatments such as radiotherapy and chemotherapy are administered depending on tumour type and stage. Watchful surveillance using clinical examination and ultrasound can be used for early-stage testicular cancer, particularly among patients who wish to postpone testicular cancer treatment for paternity purposes [5]. However, it was found that clinical examination and ultrasound are not accurate enough to include patients in surveillance protocols [6], thus increasing the risk of disease progression [5].

A unilateral painless mass is the most common symptom of testicular cancer and, in 80% of cases, is discovered accidentally by men [7, 8]. Most testicular lumps, however, are benign caused by diseases such as epididymo-orchitis (i.e., infection in the epididymis and/or the testis) [9], and varicocele (i.e., dilation of the pampiniform venous plexus) [10]. Other testicular diseases include testicular torsion (i.e., twisting of the testis) [11], which often causes severe scrotal pain and can lead to necrosis [11]. Testicular torsion requires aggressive management in men presenting with testicular pain that has been ongoing for many hours, even 24 hours or more from the onset of ischemia [11].

Raising awareness and promoting testicular self-examination and early help-seeking can help detect testicular diseases early, reduce long-term treatment costs, and improve health outcomes. Experiencing testicular symptoms, however, does not necessarily mean that men would seek help. In the past, we found that barriers to help-seeking for symptoms of testicular disease among men include lack of awareness, embarrassment, fear of a cancer diagnosis, and symptom misappraisal [12]. In addition, in our systematic reviews (n = 6), we found men's awareness of testicular diseases, intentions to seek help for symptoms of concern, and behaviours in terms of performing testicular self-examination were all suboptimal [13–17].

Athletes involved in contact sports have an increased risk of testicular trauma and subsequent diseases. A survey of 731 athletes in the USA found that the prevalence of testicular injuries in field games was 48.5% [18]. Hurling, one of the oldest and fastest field games in Ireland, is played with a ball made from cork (i.e., sliotar) covered in leather and struck with a wooden stick (i.e., hurley). When struck, the sliotar can reach a speed of 160 km/h which poses significant injury risk either by the sliotar or the hurley [19]. A single-centre Irish study reported that, out of 70 patients presenting with penoscrotal injuries, 10 (14%) had injuries caused by blunt scrotal trauma whilst playing hurling [20]. Of those, 9 (90%) had positive findings on ultrasonography, 3 (33%) required operation, and 1 (10%) underwent an orchidectomy [20].

The use of virtual reality (VR) in general [21], and in healthcare in particular [22] is on the rise. The evidence available to date to support the use of VR in men's health promotion is limited. We previously found that VR was user-friendly and acceptable for attracting younger men into research on testicular diseases [23]. We also conducted a one-group pre-post pilot study where we delivered the Enhancing Men's Awareness of Testicular Diseases using a Virtual Reality (E-MAT$_{VR}$) intervention to help raise men's awareness of testicular diseases, promote testicular self-examination, and encourage early help-seeking [24]. Findings were promising; however, a major limitation was the lack of a control group. Therefore, we were unable to determine whether the changes in outcomes were caused by the intervention, the use of repeated measures, or unforeseeable factors [24]. Moreover, to the best of our knowledge, no trials have investigated the use of VR in men's health promotion and no studies have evaluated the process of delivering VR in a trial from the perspective of participants and researchers.

Pilot feasibility studies and process evaluations are important steps in the Medical Research Council (MRC) framework for developing and evaluating complex interventions [25]. In their conceptual framework, Eldridge et al. [26] emphasised the importance of pilot feasibility studies in preparation for randomised controlled trials (RCTs). Such studies ask whether something can be done, should researchers proceed with it, and if so, how. They also assist with decision-making in preparation for a future RCT, including decisions around randomisation and recruitment [26].

Trials with no process evaluations are rarely adequate for complex intervention research [25]. Process evaluations combining qualitative and quantitative methods help answer questions around fidelity, quality of implementation, mechanisms of change, and context. Such evaluations are key to determining why an intervention fails, why it works, and how it can be optimised [25].

In the context of our study, the conduct of a pilot feasibility study with embedded mixed-method process evaluation helped us explore uncertainties that need to be addressed prior to conducting a future RCT. These include: recruitment; randomisation; the use of a comparator; participant retention; time taken to deliver the intervention; practicality of delivering the intervention; as well as intervention feasibility, usability, and adherence.

### Aim

The aim of our study was to pilot the effect of the E-MAT$_{VR}$ intervention compared to the Enhancing Men's Awareness of Testicular Diseases using Electronic information control (E-MAT$_E$) among male athletes and coaches engaged in indigenous Gaelic games. A mixed method process evaluation was embedded in the pilot feasibility study to understand and mitigate potential sources of future intervention failure and explore participants' and researchers' experiences of E-MAT$_{VR}$ and E-MAT$_E$.

## Materials and methods

### Trial design

A two-arm parallel-group randomised (1:1) pilot feasibility study was conducted in line with Eldridge et al.'s [26] guidance and conceptual framework. The trial protocol is registered in ClinicalTrials.gov (Identifier: NCT05146466) [27] and is available in S1 File. There have been no deviations from the pilot feasibility study protocol. Reporting of the trial adhered to the Consolidated Standards of Reporting Trials (CONSORT) statement for pilot and feasibility trials [28] (S2 File). Participant enrolment, allocation, follow-up, and analyses are illustrated using the CONSORT flow diagram (Fig 1).

The embedded mixed method process evaluation applied a descriptive realist evaluation using MRC guidance on process evaluations of complex interventions [29]. This involved exploring what works, for whom, and under what circumstances [30]. In compliance with MRC guidance [29], a logic model adapted from the Kellogg Foundation Framework was used to summarise the inputs, activities, outputs, outcomes, and impact of the intervention (Fig 2). The evidence-based underlying assumptions were also described along with barriers and facilitators for implementation [13, 15, 16, 20, 23, 24, 31, 32]. The protocol for the mixed method process evaluation is published elsewhere [33]. There have been no deviations from the mixed method process evaluation protocol.

### Participants

**Eligibility criteria.** Individuals eligible to participate in the study met the following criteria: (i) assigned male at birth; (ii) aged 18 to 50 years (i.e., age group at highest risk for testicular diseases); (iii) residing in the Republic of Ireland; and (iv) players and/or coaches in Gaelic Athletic Association (GAA) clubs. GAA is Ireland's largest international amateur sporting and cultural organization, with over 500,000 members in 2,200 clubs nationally [34]. GAA is focused primarily on promoting indigenous Gaelic games and pastimes, which include the traditional Irish sports of hurling, camogie, Gaelic football, Gaelic handball, and rounders [34]. Of note, given that the focus of our study was on process and feasibility rather than efficacy, individuals with a history of testicular disease were eligible for inclusion.

Individuals who were younger than 18 years or older than 50 years, not assigned male at birth, residing outside the Republic of Ireland, and not involved in GAA as either players or coaches were excluded. VR rarely causes motion sickness and the risk for seizures is negligible [35]. However, for safety reasons, individuals with a history of seizures and/or motion sickness

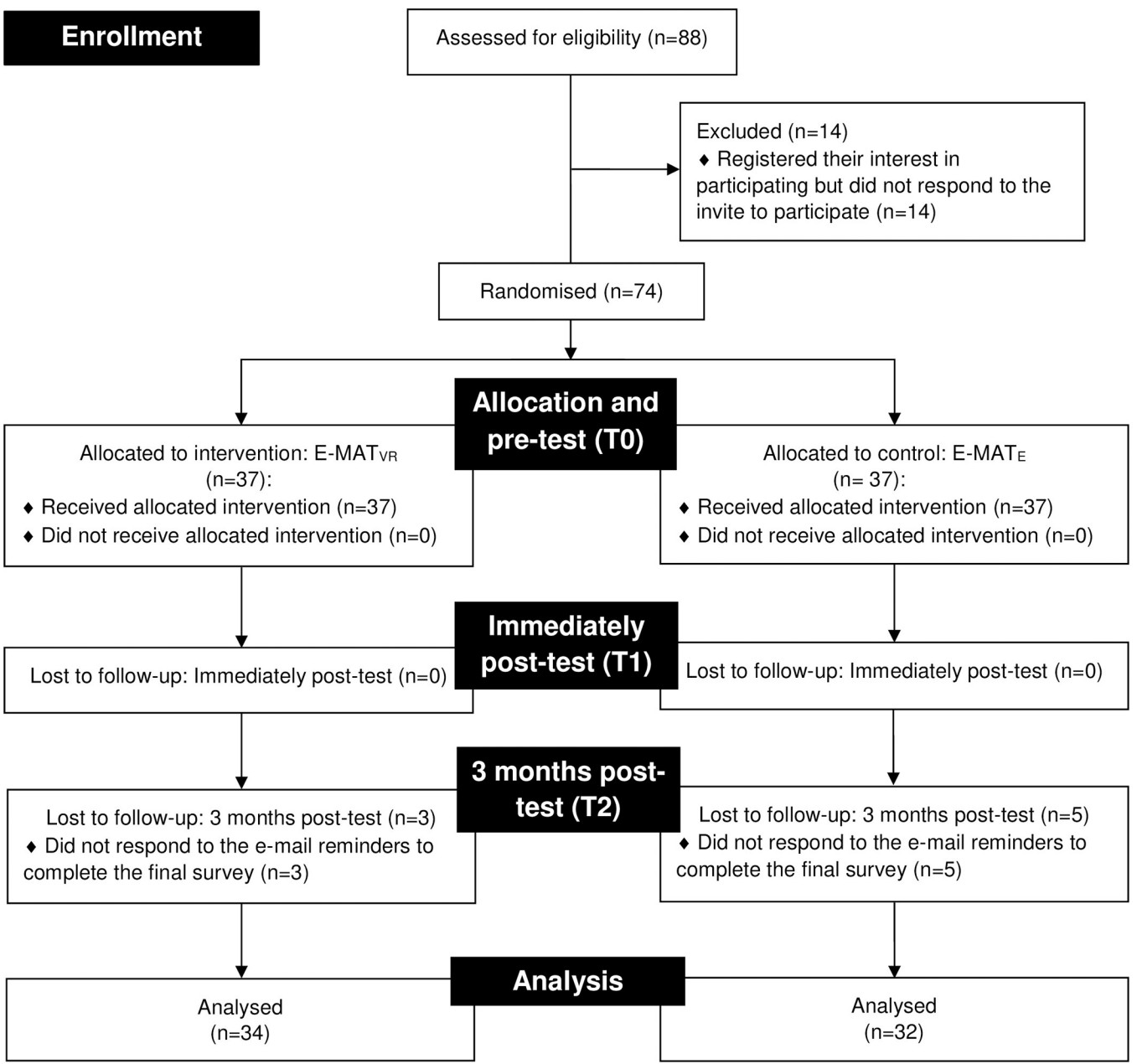

**Fig 1. CONSORT flow diagram for the E-MAT pilot feasibility study.**

were excluded. The four researchers who delivered E-MAT$_{VR}$ and E-MAT$_E$ and collected data were also eligible to participate in the process evaluation to explore their experiences conducting the study.

**Setting.**   GAA clubs are increasingly adopting "The Healthy Club" philosophy to improve the health of their members [32]. Therefore, participation was primarily sought from geographically dispersed Healthy GAA clubs in the south of the Republic of Ireland.

An invitation letter was e-mailed on two occasions to the executive committee of each club. Clubs that responded to the e-mail and expressed their interest in participating were asked to

| Mobilised resources | Activities and interventions | Specific processes to measure | Short-term outcomes and measures / Long-term outcomes reflecting program objectives |
|---|---|---|---|
| **Resources (personnel)**<br>• Male Gaelic Athletic Association (GAA) players and coaches<br>• GAA gatekeepers (Chairs, Public Relations Officers, Healthy Club Officers)<br>• Researchers who collected data<br>• Research team<br>• Funder<br>• Trial sponsor<br><br>**Resources (other)**<br>• GAA sports club facilities<br>• Social media (X™ [formerly Twitter™] and Facebook™) for recruitment.<br>• Posters with QR codes for recruitment.<br>• Equipment including virtual reality (VR) headset used to deliver E-MAT$_{VR}$ (intervention) and tablet used to deliver E-MAT$_E$ (control) and collect survey data. | Observational intervention fidelity checks | Fidelity checks during testing to measure if intervention is delivered as intended and assess the quality of intervention delivery. | Fidelity |
| | Dose and reach considerations | Dose: Time taken for intervention delivery, completion, satisfaction, and receptivity.<br>Reach: Proportion of intended population that participated measured by the number of attendees. | Dose and reach |
| | Usability and Satisfaction Survey | Usability and Satisfaction Survey exploring participants' experiences of and satisfaction with E-MAT$_{VR}$ and E-MAT$_E$. | Usability and satisfaction |
| | Individual interviews | Semi-structured individual interviews to explore participants' experiences of E-MAT$_{VR}$ and E-MAT$_E$. | Participants' experiences of E-MAT$_{VR}$ and E-MAT$_E$ |
| | Focus group | Focus group with researchers to explore experiences delivering E-MAT$_{VR}$ and E-MAT$_E$. | Researchers' experiences of delivering E-MAT$_{VR}$ and E-MAT$_E$ |

| Assumptions (root cause analyses, prior learning/experience) | External Factors (barriers/facilitators) |
|---|---|
| • Men prefer brief, innovative, interactive, visually appealing, and light-hearted health promotion interventions.<br>• Men lack awareness of testicular diseases, very few practise testicular self-examination and many do not know what to look for during testicular self-examination.<br>• Men delay seeking help and/or are unwilling to seek help for symptoms of testicular disease.<br>• Men perceive testicular diseases and testicular self-examination education as important.<br>• VR is a potentially effective health promoting platform among at-risk age groups.<br>• GAA players are particularly at risk for testicular trauma and subsequent diseases.<br>Raising awareness, promoting testicular self-examination and early help-seeking can help detect testicular diseases early, reduce treatment costs, and improve health outcomes. | **Barriers:**<br>• Access to GAA clubs<br>• Recruitment issues<br>• COVID-19 and associated restrictions<br>• Potential lack of interest in VR technology<br>• Side effect of VR technology<br>• Evolving technology and cost of technology<br>**Facilitators:**<br>• Pilot feasibility study *(reported in present paper)*.<br>• Study Within A Trial (SWAT) to determine which recruitment method(s) (X™/Facebook™/poster with quick response [QR code]) is/are most effective to recruit and retain participants.<br>• Economic evaluation to explore the cost-benefit of VR technology.<br>• Mixed-method process evaluation to describe the experiences of participants/key stakeholders and barriers and facilitators to implementation *(reported in present paper)*.<br>• Recruiting participants from geographically dispersed GAA clubs who have adopted the "Healthy Club" philosophy to improve the health of their players.<br>• Involvement of patient and public representatives in the study.<br>• Use of state-of-the-art VR technology.<br>• Incentive given at the conclusion of the trial. |

**Fig 2. Process evaluation inputs, activities, outputs, outcomes, impact, underlying assumptions, and barriers and facilitators for implementation.**

advertise the study on their Facebook[TM] and X[TM] (formerly Twitter[TM]) pages. Social media posts explained what the study involved and comprised a link that club members accessed to provide their contact details and register their interest in participating. Ten posters with a quick response (QR) code with the same link were displayed in each participating club. Each recruitment strategy (i.e., Facebook[TM], X[TM], and poster with QR code) was implemented over two weeks per club at random, for a total of six weeks [36].

Club members who registered their interest in participating were contacted by the researchers to assess their eligibility. Potential participants who were eligible were provided with an information leaflet explaining the study process and the risks and benefits from their participation and were given an opportunity to speak to a member of the research team if they had any questions or concerns. They were then asked to provide written informed consent in the presence of research personnel, and those who did were enrolled into the study. Participants were required to sign a second informed consent form if they wished to participate in qualitative interviews as part of the process evaluation. Researchers were also asked to provide written informed consent if they wished to be interviewed.

**Procedures.** Data were collected at baseline (T0), immediately post-test (T1), and three months post-test (T2) using the Castor EDC software, a secure clinical data management platform that enables researchers to easily capture and integrate data [37]. Participants completed a baseline questionnaire (T0). They were then individually randomised to either trial arms. Immediately after exposure to E-MAT$_{VR}$ or E-MAT$_E$, they completed a second questionnaire (T1). Four research personnel collected T0 and T1 data in-person in participants' respective GAA clubs, usually before a training session.

Three months post-test (T2), participants received a link by e-mail to complete the third and final questionnaire. Two subsequent reminders were sent, and non-respondents were considered lost to follow-up. Participants who completed T0, T1, and T2 questionnaires received a gift voucher as a token of appreciation.

## Interventions

The study followed MRC guidance for developing and evaluating complex interventions [25]. Participants randomised into the active arm of the study received E-MAT$_{VR}$, which was developed between 2014 and 2018. E-MAT$_{VR}$ is underpinned by the Preconscious Awareness to Action Framework [38], and the concept "testicular awareness" [39]. The development and usability testing of E-MAT$_{VR}$ is reported elsewhere [23, 24].

E-MAT$_{VR}$ is an educational intervention aimed at raising awareness of testicular diseases and promoting testicular self-examination. It is a bespoke computer software installed onto the latest VR technology and delivered using a wireless headset, handheld controllers, and built-in headphones with voiceover. E-MAT$_{VR}$ is an interactive serious gaming experience with three individual areas, representing three gaming levels. The intervention takes approximately 10 minutes to complete. A tutorial on how to use the VR headset and controllers is embedded within E-MAT$_{VR}$.

The E-MAT$_{VR}$ intervention begins with a sequence of popping words as the voiceover reads them out. These include light-hearted synonyms for the testes (e.g., balls, nuts, and gonads). Of note, the words "nuts" and "balls" were used rather than "testes" and "testicles" to help engage participants with this intimate and sensitive topic [31]. Participants were then transported to a virtual apartment with each room representing one game area. They were required to complete tasks in one area to move to the next area.

Area 1 is the shower, the ideal place to perform testicular self-examination. It involves a 3D space represented as an oversized shower with two walnuts floating in the centre. Participants

were asked by the voiceover to move around the walnuts using the handheld controllers, while providing information about the normal size and shape of the testes. Three changes (i.e., lump, swelling, and pain [represented as flashing light]) then appeared consecutively. These were associated with humorous responses from the voiceover. The participant was required to find all three changes to move to the next area.

Area 2 is the bedroom. It involves using a 3D model of a testis. The spermatic cord, epididymis, and lump/tumour are represented in this model. In this area, the voiceover connects symptoms experienced in area 1 to testicular structures. For example, the spermatic cord glows to indicate testicular torsion and a purple lump appears, indicating a growth/lump. Participants were required to click on each of the three structures to move to the next and final area.

Area 3 is the kitchen living room. Here, key messages are reiterated by the voiceover using three objects: (i) a poster of a fingerprint to remind participants that their testes are unique, hence the importance of knowing what is normal for them; (ii) an infographic with the steps for testicular self-examination; and (iii) a first aid kit to prompt participants to seek help for symptoms and seek emergency/immediate medical attention for sudden and severe testicular pain. Participants were required to click on each of the three objects to hear the messages. Screenshots from the different areas within the E-MAT$_{VR}$ intervention are presented in Fig 3.

Participants in the control arm (E-MAT$_E$) received the same information as E-MAT$_{VR}$ but delivered as plain text in Portable Document Format (PDF), with screenshots from E-MAT$_{VR}$ similar to those in Fig 3. E-MAT$_E$ participants were given up to 10 minutes to read the text and look at images using a tablet.

## Outcomes

The full details of the instruments used to collect data in the pilot feasibility trial and process evaluation including how and when they were used are presented in Table 1. All instruments

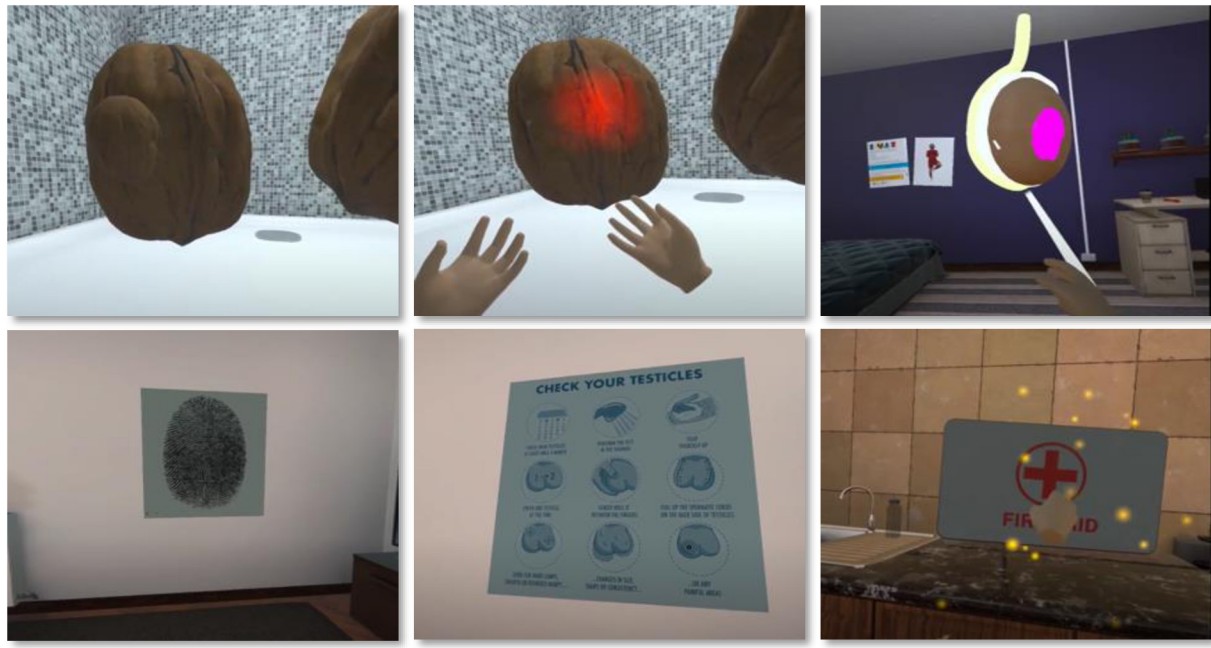

**Fig 3. Screenshots from the different areas within the E-MAT$_{VR}$ intervention.**

used in the pilot feasibility trial and most instruments used in the process evaluation were developed, validated, and pilot tested elsewhere [24, 40–42]. No changes were made to these instruments in the current study.

## Process evaluation outcomes

Participants who consented were interviewed individually either in-person immediately after T1 or later via telephone. The research team participated in a virtual focus group to glean their perspectives of delivering the intervention/control. A semi-structured interview guide was used for participants and researchers (S1 Table in S3 File). Interview guides were developed by four senior academics and researchers who have experience in qualitative research (MMS, MMC, MPD, JH). The interviews and focus group explored how E-MAT$_{VR}$ and E-MAT$_E$ were perceived by participants and researchers, if/why this varied, and how these perceptions affected intervention/control receptivity. The interviews and focus group also explored the mechanisms through which the intervention/control brought about changes in testicular awareness, which is crucial to understanding how the effects of the intervention/control occurred and how these effects might be replicated in the future [30].

Context which includes anything peripheral to the intervention/control that may influence its implementation, reach, or effects was measured using a survey. The interview and focus group topic guide explored in what context the intervention/control was effective or ineffective, and whether they might be transferrable to other contexts. The usability and satisfaction survey contained a checklist of items with three of those items examining the impact of context on the implementation of the intervention/control. This included determining whether the intervention/control was applicable to real life, men across a range of ages (18 to 50 years), and men from different ethnic backgrounds.

Fidelity checks were conducted to check if E-MAT$_{VR}$ and E-MAT$_E$ were delivered as intended. This involved having a researcher observe participants during testing and recording (tick box) pre-specified items (e.g., participants' behaviour, interest, and engagement) [43]. The quality of delivery was also assessed during the fidelity checks. These checklists were purposefully developed for use in the current study. Checklist items were mapped onto the processes and contents of the E-MAT$_{VR}$ intervention and E-MAT$_E$ control. These checklists were pilot tested with four participants. Data from these participants were not included in the final analysis. No changes were made to these checklists following pilot testing. Data were collected on dose exposure, dose completeness, and dose satisfaction. The time taken to deliver E-MAT$_{VR}$ and E-MAT$_E$ and the units of intervention/control completed (dose completeness) were captured in minutes. Data were recorded automatically within and extracted from the VR headset for E-MAT$_{VR}$ participants. The time it took participants to complete E-MAT$_E$ in minutes was recorded manually by the observing researcher. Participants' satisfaction with the intervention/control, interactions with research staff (dose satisfaction), and extent to which participants were receptive to the intervention/control (dose exposure) were collected in the usability and satisfaction survey.

Reach refers to the proportion of the intended population that participated in the study measured by the number of participants within each arm [29]. In our study, reach was determined by comparing the number of participants who consented to participate relevant to the number of GAA players and coaches who registered their interest in participating but did not.

Feasibility and satisfaction were measured using instruments which assessed usefulness, ease of use, and level of satisfaction with use of both E-MAT$_{VR}$ and E-MAT$_E$ [23, 42].

Finally, four open-ended questions explored elements of the study that worked and did not work for participants, and any possible changes they would like to make [23, 42]. Process

**Table 1. Data collection instruments used in the feasibility trial and process evaluation.**

| Instrument* | Source | Number of items | Time administered | Answer options | Scoring | Interpretation of scores |
|---|---|---|---|---|---|---|
| **Pilot feasibility trial** | | | | | | |
| Sociodemographic Questionnaire | [24] | 11 | T0 | 9 Multiple choice 2 Free text | N/A | N/A |
| Testicular Knowledge Questionnaire | [24] | 12 | T0, T1, T2 | Multiple choice. Each question had one correct answer. | 0–1 | Higher scores indicate greater testicular knowledge |
| Testicular Self-Examination Behaviours | [24] | 4 | 3 items at T0 1 item at T2 | Multiple choice (Yes/No) | Individual items | Yes: Performed the behaviour No: Did not perform the behaviour |
| Testicular Awareness Scale | [24] | 5 | T0, T1, T2 | Level of agreement assessed on a 5-point Likert scale | 1–5 | Higher scores indicate greater testicular awareness |
| Perceived Risk Item | [24] | 1 | T0, T1, T2 | Level of agreement assessed on a 5-point Likert scale | 1–5 | Higher scores indicate greater perceived risk |
| General Help-Seeking Questionnaire | [24, 40] | 3 | T0, T1, T2 | Likelihood to seek help assessed on a 7-point Likert scale | 1–7 | Higher scores indicate greater intentions to seek help |
| Implementation Intentions Scale | [24, 41] | 3 | T0, T1, T2 | Level of agreement assessed on a 5-point Likert scale | 1–5 | Higher scores indicate greater implementation intentions |
| Recommendation of Testicular Self-Examination to Others | [24] | 1 | T2 | Multiple choice (Yes/No) | Individual items | Yes: Performed the behaviour No: Did not perform the behaviour |
| **Process evaluation (quantitative data)** | | | | | | |
| Feasibility Survey | [23] | 9 | T1 | Level of agreement assessed on a 5-point Likert scale | 1–5 | Higher scores indicate greater feasibility |
| Satisfaction Survey | [42] | 3 | T1 | Level of satisfaction assessed on a 5-point Likert scale | 1–5 | Higher scores indicate greater satisfaction |
| E-MAT$_{VR}$ Fidelity Checklist | Researcher-designed | 6 | During testing | Five items assessed level of completion (completed, partially completed, not completed) and one item recorded reason (s) for termination using multiple choice (Yes done as planned/ No not done as planned) | 0%-100% | 80%-100%: High quality 65%-79%: Moderate quality 50%-64%: Low quality 0%-49%: Very low quality |
| E-MAT$_{E}$ Fidelity Checklist | Researcher-designed | 3 | During testing | Two items assessed level of completion (yes, partially, no) and one item captured intervention completion time (Yes done as planned/ No not done as planned) | 0%-100% | 80%-100%: High quality 65%-79%: Moderate quality 50%-64%: Low quality 0%-49%: Very low quality |
| Virtual Reality Headset | Researcher-designed | 2 | During testing | N/A | N/A | Time it took to complete E-MAT$_{VR}$ and number of participants who completed E-MAT$_{VR}$ |
| Tablet | Researcher-designed | 2 | During testing | N/A | N/A | Time it took to complete E-MAT$_{E}$ and number of participants who completed E-MAT$_{E}$ |
| **Process evaluation (qualitative data)** | | | | | | |
| Open-ended survey questions | [42] | 4 | T1 | N/A | N/A | N/A |
| Individual interviews with participants | Researcher-designed | 7 semi-structured questions | After T1 | N/A | N/A | N/A |
| Focus group with researchers | Researcher-designed | 10 semi-structured questions | After T2 | N/A | N/A | N/A |

* The same instruments were used for E-MAT$_{VR}$ and E-MAT$_{E}$, unless otherwise indicated.

N/A = not applicable; T0 = time 0 (pre-test); T1 = time 1 (immediately post-test); T2 = time 2 (three months post-test).

evaluation data including timing, difficulties experienced during E-MAT$_{VR}$/E-MAT$_E$, frequency of adverse events, and implementation problems were also collected by researchers during the study. Implementation problems identified during the trial were categorised as they arose, including the mechanisms by which the issues were addressed.

**Pilot efficacy outcomes.** The primary efficacy outcomes were testicular knowledge and testicular self-examination behaviours. The testicular knowledge questionnaire measured participants' knowledge of the anatomy of the testes, common testicular symptoms and diseases, and testicular self-examination using 12 multiple choice questions [24].

At T0, three items explored whether participants performed testicular self-examination within the past year and month [24]. They were also asked if a healthcare professional ever examined their testes. At T2, one item explored whether participants performed testicular self-examination over the past three months (i.e., since exposure to the intervention/control). All item responses were dichotomous (Yes/No).

The secondary efficacy outcomes included testicular awareness, perceived risk of testicular diseases, help-seeking intentions, implementation intentions, and recommendation of testicular self-examination to others. The testicular awareness scale is a 5-point Likert scale that assessed participants' testicular awareness, that is their familiarity with their own testes, knowledge of what is normal and what is not normal, and ability to differentiate between what is normal and what is not normal [24, 39].

Participants' perceived risk of testicular diseases was assessed using one item. The level of agreement for this item was measured on a 5-point Likert scale [24].

Participants' intentions to seek help for testicular symptoms were assessed using the General Help Seeking Questionnaire [24, 40]. Here, three items assessed participants' intentions to seek help for three different symptoms (i.e., testicular swelling, lumpiness, and pain) from various sources of help.

Implementation intentions "specify the when, where, and how of responses leading to goal attainment" (p.121) [41]. The Implementation Intentions Scale comprised three items assessing participants' intentions to feel their own testes in the shower/bath at least once over the coming month and their intentions to advise at least one man to do the same [24]. The level of agreement for each item was measured on a 5-point Likert scale.

One item assessed men's behaviours in relation to advising at least one man to feel their testes at T2. Item responses were dichotomous (Yes/No) [24].

A sociodemographic questionnaire was administered at T0. It comprised 11 items as follows: age; gender; nationality; marital status; name of GAA club; role in GAA club; highest level of education; current occupation; personal history of testicular disease(s); prior experience with VR; and where participants heard about the feasibility trial. Of note, the names of participating GAA clubs are not disclosed in this paper for confidentiality purposes.

## Sample size

The goal of a pilot feasibility study is to identify problems that would impede the conduct of a larger, definitive RCT. There is no gold standard for sample size calculation in pilot feasibility studies and sample sizes as small as 10 and as large as 59 have been recommended [24]. Julious [44], for example, suggested a sample size of 12 per arm, whereas Viechtbauer et al. [45] proposed a formula to calculate the sample size in pilot studies and reported that "if a problem exists with 5% probability in a potential study participant, the problem will almost certainly be identified (with 95% confidence) in a pilot study including 59 participants" (p. 1375). Therefore, we set the sample size in our study at 59 [45]. Allowing for a potential attrition rate of 25%, we aimed to recruit 74 participants.

## Randomisation

For each participating GAA club, individual participants were randomised to one of the two trial arms (E-MAT$_{VR}$ or E-MAT$_E$) using the Castor EDC software. Allocation concealment was maintained using the Castor EDC software. Once participants had consented and their baseline assessment (T0) had been entered into the Castor EDC software, their allocation was given by automated e-mail. Randomisation and allocation concealment were conducted according to the standard operating procedures of the Statistics and Data Analysis Unit of the Health Research Board Clinical Research Facility, University College Cork, Ireland.

Given the differences between the two arms, participants were made aware of the arm they were randomised to once they were assigned to either E-MAT$_{VR}$ or E-MAT$_E$. Outcome assessments were self-reported and returned to research staff with no information on allocation. The data analyst was also blinded to allocation until database lock and completion of the protocol-specified data analysis.

## Statistical methods

**Quantitative data analysis.** Prior to analysis, data underwent extensive quality checking. Given the feasibility aims of the trial, analyses of participant outcomes were largely descriptive. However, we did estimate between-arm contrasts for the knowledge scores and the testicular self-examination behaviours on an intent-to-treat basis. For the former, we estimated the between-arm difference in mean scores using linear regression with adjustment for baseline scores; for the latter we estimated odds ratios using logistic regression. Resulting effect estimates were reported with 95% confidence interval (CI) and p-values. Missing data were addressed using complete case analysis, following from the small number of missing values. Because the pilot feasibility trial was not designed to return a high-powered test of the efficacy of the intervention, we report these results on a purely exploratory basis, and for the purposes of informing the design of a subsequent RCT. Simple between-arm tests for ordinal responses were conducted using Kruskal-Wallis rank sum tests. For multi-item scales, we evaluated the internal consistency of items using Cronbach's alpha [46], or the equivalent Kuder-Richardson 20 (KR-20) metric [47] when the items were binary. All reported p-values are based on two-sided tests with statistical significance defined as $p < 0.05$.

Analyses were conducted using R (v 4.2.0) and RStudio (2023.06.0 Build 421). All analysis steps, including cleaning or modifications to study database, were fully scripted and replicable. Upon completion, the study database was prepared according to the findability, accessibility, interoperability, and reusability (FAIR) data principles and made available along-side the analysis scripts via the Open Science Foundation (https://osf.io/m5wb7/).

**Qualitative data analysis.** Qualitative data from the interviews with participants and focus group with researchers were analysed using qualitative content analysis with pre-defined main categories reflective of the key constructs within the logic model, with an openness to emergent themes [48, 49]. All qualitative data were coded and analysed in NVivo. Similar codes were grouped to form sub-categories which were then collated under main categories. Qualitative data analysis was conducted by one author (MMC) and checked for accuracy by two authors (MPD, JH) to minimise errors and improve data credibility and confirmability [50].

Data from the four open-ended questions were analysed using quantitative content analysis which involved systematically categorising and recording textual material so that they can be analysed [51].

### Ethical considerations

This study was conducted in line with ethical principles depicted in the Declaration of Helsinki. Ethical approval was received from the Clinical Research Ethics Committee at University College Cork (ECM 06/2023 PUB). All participants completed written informed consent to participate.

## Results

### Participants

Fourteen GAA clubs were invited to participate. Of those, nine responded and agreed to data collection in their clubs. In total, 88 individuals from the nine GAA clubs, registered their interest in participating via Facebook[TM], X[TM], and posters with QR code. Of those, 74 consented, were randomised, and completed T0 and T1 measures. The first participant consented March 15, 2022, and the last participant consented August 11, 2022. There were eight participants lost to follow up at T2, yielding a sample size of 66 participants at T2 (89.2% retention rate). T0, T1, and T2 data were collected between March and November 2022 by four trained research personnel who were not known to participants.

Participants' median age was 26 (Interquartile Range [IQR] 22 to 33). Most participants were GAA players (n = 59, 79.7%), Irish (n = 73, 98.6%), single (n = 40, 54.1%), employed (n = 49, 66.2%), university degree holders (n = 63, 85.1%), and first time VR users (n = 54, 73.0%). Two participants (2.7%) reported having a history of testicular cancer. More than half of the participants reported being referred to the study by a friend (n = 38, 51.4%). Only one participant (1.4%) learned about the study from the poster with QR code. Participants in both arms were broadly similar to each other. Sample characteristics are presented in Table 2.

All participants were invited to participate in an individual interview with one of the authors (MMC) as part of the process evaluation. Of those, nine agreed to participate (six from E-MAT$_E$ and three from E-MAT$_{VR}$). Eight interviews were conducted face-to-face and one by telephone. Interviews with participants lasted on average five minutes. Finally, all four researchers who collected data participated in a one-hour virtual focus group that was facilitated by the lead author (MMS).

### Process evaluation outcomes

**Context.** All participants in E-MAT$_{VR}$ and 97.3% (n = 36) of those in E-MAT$_E$ either agreed or strongly agreed that the intervention was applicable to men aged between 18 and 50 years. There were similar findings regarding applicability to those from different ethnic and cultural backgrounds; the majority of E-MAT$_{VR}$ (n = 33, 89.2%) and E-MAT$_E$ (n = 36, 97.3%) participants either agreed or strongly agreed. Thirty-six (97.3%) participants in both E-MAT$_{VR}$ and E-MAT$_E$ either agreed or strongly agreed that the intervention was close and applicable to real life (S2 Table in S3 File).

**Fidelity.** High quality of delivery was associated with both E-MAT$_{VR}$ and E-MAT$_E$ (fidelity score ≥80%). All participants in E-MAT$_{VR}$ received a safety briefing and an explanation on the use of the VR headset and controllers. All participants used the VR equipment and began the VR experience as instructed. Two participants in E-MAT$_{VR}$ (5.4%) ended the VR experience early. One due to nausea and one due to a technical problem in the VR headset. All E-MAT$_E$ participants could use the tablet (n = 37, 100%) and almost all E-MAT$_E$ participants appeared to read everything on the tablet (n = 35, 94.6%).

**Dose and reach.** Data on dose exposure were not available for seven participants in E-MAT$_{VR}$ and one participant in E-MAT$_E$. The mean completion time was 9.45 minutes

**Table 2. Sample characteristics for the E-MAT pilot feasibility study.**

| Characteristic | Overall N = 74* | E-MAT$_E$ N = 37* | E-MAT$_{VR}$ N = 37* |
|---|---|---|---|
| **Age in years**** | 26 (22, 33) | 29 (24, 38) | 24 (21, 28) |
| **Role in the club** | | | |
| Player | 59 (79.7%) | 25 (67.6%) | 34 (91.9%) |
| Coach | 11 (14.9%) | 9 (24.3%) | 2 (5.4%) |
| Other | 4 (5.4%) | 3 (8.1%) | 1 (2.7%) |
| **Nationality** | | | |
| Irish | 73 (98.6%) | 36 (97.3%) | 37 (100%) |
| Other | 1 (1.4%) | 1 (2.7%) | 0 (0%) |
| **Marital status** | | | |
| Single | 40 (54.1%) | 17 (45.9%) | 23 (62.2%) |
| In a relationship | 19 (25.7%) | 8 (21.6%) | 11 (29.7%) |
| Married | 15 (20.3%) | 12 (32.4%) | 3 (8.1%) |
| **Highest level of education** | | | |
| University | 63 (85.1%) | 33 (89.2%) | 30 (81.1%) |
| Highschool | 10 (13.5%) | 3 (8.1%) | 7 (18.9%) |
| Other | 1 (1.4%) | 1 (2.7%) | 0 (0%) |
| **Occupation** | | | |
| Employed | 49 (66.2%) | 29 (78.4%) | 20 (54.1%) |
| Student | 24 (32.4%) | 7 (18.9%) | 17 (45.9%) |
| Self-employed | 1 (1.4%) | 1 (2.7%) | 0 (0%) |
| **Personal history of testicular disease** | | | |
| No | 71 (95.9%) | 36 (97.3%) | 35 (94.6%) |
| Yes | 2 (2.7%) | 1 (2.7%) | 1 (2.7%) |
| Unsure | 1 (1.4%) | 0 (0%) | 1 (2.7%) |
| **Previous use of virtual reality** | | | |
| No | 54 (73.0%) | 30 (81.1%) | 24 (64.9%) |
| Yes | 20 (27.0%) | 7 (18.9%) | 13 (35.1%) |
| **Where participants heard about the study** | | | |
| Friend | 38 (51.4%) | 18 (48.7%) | 20 (54.1%) |
| Facebook[TM] | 10 (13.5%) | 7 (18.9%) | 3 (8.1%) |
| X[TM] | 6 (8.1%) | 2 (5.4%) | 4 (10.8%) |
| QR code | 1 (1.4%) | 1 (2.7%) | 0 |
| Friend/Facebook[TM]/X[TM] link via WhatsApp[TM] | 14 (18.9%) | 9 (24.3%) | 5 (13.5%) |
| Other | 5 (6.8%) | 0 | 5 (13.5%) |

* n (%)

** Median (IQR)

(range 7.15 to 12.31 minutes) for E-MAT$_{VR}$ and 4.07 minutes (range 1.50 to 7 minutes) for E-MAT$_E$. As for reach, the target sample size of 74 participants was reached.

**Usability and satisfaction.** Most participants (67/74, 90.5%) either agreed or strongly agreed that the device was comfortable to use in both E-MAT$_{VR}$ (n = 34, 91.9%) and E-MAT$_E$ (n = 33, 89.2%). All participants in E-MAT$_{VR}$ agreed or strongly agreed that the device was easy to use and that they were engaged to learn by the device. While 33/37 E-MAT$_E$ participants (89.2%) agreed or strongly agreed with these statements. Similarly, 34/37 E-MAT$_{VR}$ participants (91.9%) either agreed or strongly agreed that the time it took to complete was reasonable and all E-MAT$_E$ participants agreed with this statement. All 74 participants either

agreed or strongly agreed that the information provided by the device was clear (S2 Table in S3 File).

All 74 participants were either extremely satisfied or somewhat satisfied with their experience of using the PDF reader on the tablet (E-MAT$_E$) or VR system (E-MAT$_{VR}$) to learn about testicular diseases. Similarly, all E-MAT$_{VR}$ participants and 36/37 E-MAT$_E$ participants (97.3%) were either extremely satisfied or somewhat satisfied with their overall experience of participating in the study (S3 Table in S3 File).

**Open-ended questions and qualitative interviews with participants.** All 74 participants answered the four open-ended questions at T1. The full answers are presented in S4 Table in S3 File. Participants in E-MAT$_{VR}$ (n = 37) described the VR game as *"interactive"* (n = 16, 43.2%), *"easy"* (n = 13, 35.1%), *"fun"* (n = 7, 18.9%) and close to *"real life"* (n = 4, 10.8%). Elements of the VR game that participants did not favour were the risk of *"nausea"* and *"sickness"* (n = 6, 16.2%*)*, as well as *"dizziness"* (n = 5, 13.5%) and difficulties with *"navigation"* (n = 4, 10.8%). Most E-MAT$_{VR}$ participants (n = 30, 81.1%) did not recommend making any changes to the VR game. Recommended changes included having *"more interaction"* (n = 2, 5.4%) and *"more content"* (n = 1, 2.7%).

Participants in E-MAT$_E$ (n = 37) mainly liked the tablet because it was *"easy"* (n = 22, 59.5%), *"quick"* (n = 6, 16.2%), contained *"pictures"* (n = 4, 10.8%), and *"informative"* (n = 3, 8.1%). The main elements of the tablet not favoured related to the *"touchscreen function"* (n = 7, 18.9%). Others reported *"repetition"* (n = 1, 2.7%) and *"lack of interaction"* (n = 1, 2.7%) as drawbacks of E-MAT$_E$. Most participants in E-MAT$_E$ (n = 24, 64.9%) responded with *"none/N/A"* when asked about elements of the tablet that they recommend changing.

Responses were explored in greater depth during qualitative interviews with nine participants. As a result, seven main categories and 22 sub-categories were identified. The seven main categories were: Positive aspects of the study; positive aspects of E-MAT$_{VR}$; positive aspects of E-MAT$_E$; challenging aspects of E-MAT$_{VR}$; challenging aspects of E-MAT$_E$; motivation for participating; and recommendations for future research. The main categories, sub-categories, and codes are presented in Table 3. The letter "P" is used to designate participants hereafter.

*Positive aspects of the study.* Participants in both, E-MAT$_{VR}$ and E-MAT$_E$ described their experience of participating in the study as *"enjoyable"* (P2, P3, P9) *"easy"* (P2, P3, P4) and *"informative"* (P3). One participant seemed motivated to perform testicular self-examination following his participation in the study: *"It brings it more to the front of your mind to do checks on yourself"* (P8).

*Positive aspects of E-MAT$_{VR}$.* VR technology was perceived as *"engaging"* (P4, P6), *"interactive"* (P3, P9), *"fun"* (P3, P4), and *"quick to use"* (P4). Although E-MAT$_E$ was described as *"more efficient"* (P9), E-MAT$_{VR}$ was perceived as *"more interactive"* (P9). As a result, participants felt that E-MAT$_{VR}$ was *"much better than writing on a piece of paper"* (P5). E-MAT$_{VR}$ was also perceived as a *"better way of learning"'* (P5) and a *"different type of learning"* (P4), encouraging self-directed learning: *"you were in control of your learning"* (P3).

*Positive aspects of E-MAT$_E$.* Many participants described E-MAT$_E$ as *"easy to use"* (P1, P6, P7, P8, P9), *"handy"* (P1, P8) and *"straight forward"* (P8). E-MAT$_E$ was also perceived as *"informative'* (P6), *"educational"* (P8), written in *"simple language"* (P2) and presented *"in a simple format"* (P8) making it *"simple for a layman to understand"* (P8). In addition, the use of the tablet during E-MAT$_E$ enabled participants to *"go through the slides at your own speed, your own pace..."* (P6).

*Challenging aspects of E-MAT$_{VR}$.* For some participants, navigating the VR environment proved challenging initially as the *"controllers were a bit confusing at first"* (P4). One participant reported *"feeling slightly dizzy at the start"* (P4). Fear of the unknown was also an initial

concern voiced by one participant: *"I felt a tiny bit anxious at the very beginning. . .because you don't know"* (P5). A technical issue was reported by another participant: *"It* [E-MAT$_{VR}$] *froze at the end"* (P3).

**Table 3. Findings from qualitative interviews with participants on their views of E-MAT.**

| Main categories | Sub-categories | Codes |
|---|---|---|
| **Positive aspects of the study** | Enjoyable | • Enjoyable experience |
| | | • Enjoyed participating |
| | Informative | • Study reminds you to check yourself |
| | | • Good to highlight issue so you will be more likely to check |
| | | • Very informative |
| | | • Learned new vocabulary and statistics relating to the epididymis |
| | Easy | • Easy to do |
| | | • Easy to understand |
| Positive aspects of E-MAT$_{VR}$ | Interactive and engaging technology | • Engaging and enjoyable |
| | | • Interactive and hands on |
| | | • Engaging and a novelty |
| | Quick to use | • Quick |
| | Fun to use | • Fun |
| | Fostering a different and better way of learning | • VR allows own control of learning |
| | | • VR is a better way of learning |
| | | • VR is a different type of learning |
| Positive aspects of E-MAT$_E$ | Ease of use | • Tablet was easy to read |
| | | • Tablet was easy to see diagrams |
| | | • Slides were simple to understand |
| | Clear and informative | • Tablet content was educational |
| | | • Tablet was informative |
| | | • Slides included educational diagrams and explanations |
| Challenging aspects of E-MAT$_{VR}$ | Getting used to navigating the VR environment | • Controllers were confusing at start |
| | | • Anxious at the beginning |
| | | • Dizzy at start of VR |
| | Nausea | • Nausea |
| | Practical issues | • Not practical for a large number of people |
| | Technological issues | • VR game froze |
| | | • Face masks lead to headset fogging up |
| | | • Tricky having to use personal Facebook™ accounts |
| Challenging aspects of E-MAT$_E$ | Lack of interactivity and engagement | • Tablet was interactive to a certain degree but more videos needed |
| | | • Tablet needs more visual content |
| | | • Missing interactivity on tablet |
| | Information overload | • A lot of slides with too much text |
| | | • Hard to take all the information in |
| **Motivation for participating** | Learning about testicular diseases | • Participated to learn about men's health |
| | | • Participated to learn about the disease |
| | | • Never previously learned about the disease other than a small bit in secondary school |
| | Quick, not time consuming and easy to do | • Participated because study was quick/not an inconvenience |
| | | • Participated because study was simple and not time consuming |
| | Voucher/ Money | • 50-euro voucher as an incentive to participate |

*(Continued)*

**Table 3.** (Continued)

| Main categories | Sub-categories | Codes |
|---|---|---|
| **Recommendations for future research** | Edit and shorten survey questions | • Prefer shorter survey/less questions |
| | Incorporate VR into education and learning | • Introduce VR in secondary schools in older age groups |
| | | • Recommend learning via tablet as was straight forward |
| | | • Recommend using a tablet because they are easy to follow |
| | More interactivity and visual content | • Make tablet more interactive and engaging |
| | | • More slides with videos |
| | | • More slides with visual interaction |
| | Target male environments | • Target male dominated areas |
| | | • Roll out study in a lot of GAA clubs |

*Challenging aspects of E-MAT$_E$.* Participants felt that E-MAT$_E$ *"wasn't very interactive"* (P2) or interactive only *"to a certain degree"* (P7). Some participants reported that E-MAT$_E$ had *"too much text"* (P8) and a *"good bit of information"* (P7). As a result, one participant reported information overload: *"Hard to take it all in"* (P7).

*Motivation for participating.* Some participants were motivated to participate in the study to *"find out more about men's health"* (P9), particularly in relation to testicular diseases, as described by one participant: *"I didn't really have too much information on the disease, only a small bit there in secondary school"* (P7). Others volunteered to participate because the study was *"easy"* (P6), *"quick"* (P8), *"simple"* (P9), and *"wasn't time consuming"* (P9). For many, the *"50-euro voucher"* (P6, P7, P8, P9) given to participants following completion of the third and final questionnaire at T2 incentivised them to participate.

*Recommendations for future research.* Participants made several recommendations to improve E-MAT$_{VR}$ and E-MAT$_E$. Both, E-MAT$_{VR}$ and E-MAT$_E$ participants recommended making the VR game and tablet *"more interactive"* (P5, P8). Several E-MAT$_E$ participants advised including *"videos"* (P6, P7, P8) and *"visual interaction"* (P7). One participant recommended incorporating VR into secondary school and university curricula: *"Maybe it's something that should be introduced in secondary schools, colleges, as part of a curriculum. . .to show it to older classrooms, older age groups in secondary schools"* (P6). Others recommended using the tablet because it is *"straightforward"* (P8), *"easy to follow"* (P6), and *"people are well used to using the device"* (P7).

**Focus group with researchers.** Five main categories and 21 sub-categories were identified from the focus group with the four researchers who collected data. The five main categories were: Positive aspects of E-MAT$_{VR}$; challenges to engaging with E-MAT$_{VR}$; resources incurred; surveys at T0 and T1; and recommendations for future research. The main categories, sub-categories, and codes are presented in Table 4. The letter "R" is used to designate researchers hereafter.

*Positive aspects of E-MAT$_{VR}$.* Three research personnel discussed the benefits of using VR in various contexts and settings (R1, R3, R4). One researcher discussed the benefits of using VR in education: *"So yes across education, it could be provided in any context really"* (R1), while another spoke about the value of using VR in other health contexts: *"So I know as well you have a student working a breast cancer one. . .you can do anything with it technically"* (R4). In addition, researchers discussed the *"enjoyment"* (R3) element of VR which made it a *"fun way for them* [participants] *to learn"* (R1). One researcher felt E-MAT$_{VR}$ was valuable for educating students prior to going on clinical placements: *". . .they* [participants] *get that sense of what it might be like before they go on placement. . .that experience and that practice beforehand–I think it could be really helpful"* (R1). Another researcher felt that E-MAT$_{VR}$ served as a *"humorous*

**Table 4. Findings from the focus group with researchers.**

| Main categories | Sub-categories | Codes |
|---|---|---|
| Positive aspects of E-MAT$_{VR}$ | Can be used in various contexts and settings | • VR can do anything and just depends on how you want to present the information |
| | | • VR is helpful across education and in any context |
| | | • VR is interactive and real-life experience so you can put it in any context |
| | Enjoyable and fun | • Participants enjoyed new experience |
| | | • Fun and innovative way to learn |
| | | • Fun way to learn |
| | Informative, helpful, and educational | • VR is informative and a memorable learning experience |
| | | • VR is helpful experience for students prior to placement |
| | | • VR is helpful in educating students |
| | | • VR helps with learning process |
| | Humorous delivery of information | • Good and a humorous way to give information |
| | | • Light humour to educate |
| | | • Humours information kept people interested |
| | Interactive and interesting technology | • VR is more interesting and interactive for men |
| | | • More boring to learn via PDF |
| Challenges to engaging with E-MAT$_{VR}$ | Delivery of information | • One participant ended game early |
| | | • If information is delivered poorly, participants can get distracted |
| | | • Some participants struggled in tutorial |
| | Nausea | • One person was nauseated |
| | Technological issues | • Ensuring VR headsets batteries are charged |
| | | • Face masks lead to headset fogging up |
| | | • Tricky having to use personal Facebook account |
| **Resources incurred** | Personnel | • Helpful to have other researchers available for support and expertise |
| | | • Personnel helped run things smoothly |
| | | • Too busy to do process evaluation interviews |
| | | • Too busy administering surveys to do fidelity checks |
| | | • Lack of manpower when participants arrive late or early |
| | Time | • Additional time and work for delivering E-MAT$_{VR}$ |
| | | • Allocating time and research members to conduct process evaluation interviews |
| | | • No time to conduct process evaluation interviews |
| | Recruiting participants | • Recruitment- one-hour slots were daunting |
| | | • Public representative helped encourage uptake in clubs |
| | | • Interest drop once participants are emailed |
| | | • Effective to remind participants of voucher |
| | Access to GAA clubs | • GAA access (onsite for data collection) was tricky so learned to collect at later times |
| | | • Hurdle in getting responses from clubs to set up time and place for data collection |
| | | • Hurdle in getting Public Relation Officers to post on social media |
| **Surveys at T0 and T1** | Similar and overlapping questions | • Participants were frustrated with questions in survey that were similar (pre- and post-test) |
| | | • Participants had to answer overlapping and similar questions (pre- and post-test) |
| | | • Questions were very similar |
| | No technical and usability feedback | • No technical feedback from the surveys |
| | | • No questions about usability of the software in the surveys |

*(Continued)*

**Table 4.** (Continued)

| Main categories | Sub-categories | Codes |
|---|---|---|
| **Recommendations for future research** | Make VR accessible | • Need to bring interventions to people as they can't download (as it is a big file) |
| | | • VR should be available where people can go and use it freely |
| | | • VR should be open to all sports players and not just GAA |
| | Incorporate VR into education and learning | • Bring VR to schools for GAA clubs to show children |
| | | • VR should be somewhere where people can learn in their own time |
| | Accommodate different levels of VR experience | • Longer tutorials for those who struggle with VR |
| | | • Ask participants about their level of experience with VR |
| | | • A process for dealing with those who are not used to VR |
| | | • Someone experienced with VR to help with technical things during data collection |
| | Training for staff | • Streamline Castor EDC software process |
| | | • More training and tutorials for Castor EDC software |
| | Usability testing | • More user testing at the start |
| | | • More testing to help find bugs |
| | Edit/shorten survey questions | • Frame questions differently to get more insight |
| | | • Remove overlapping questions to get more insightful answers |
| | | • Explain to participants that the questions are similar in pre- and post-test surveys |
| | Tailored recruitment strategy | • (Going forward) look at a more detailed recruitment strategy |
| | | • Get agreement with organisations in future |
| | | • Keep language informal to make it more appealing to participate |

*way to give the information"* (R4) which might help *"keep people interested"* (R4) and educated about testicular diseases. In contrast to E-MAT$_E$ which was described by one researcher as *"more boring"* (R1), E-MAT$_{VR}$ was described as *"more interesting and interactive"* (R2).

*Challenges to engaging with E-MAT$_{VR}$.* Information delivery proved challenging, as described by one researcher: *"They [participants] are really just being told to point at something"* (R4). Technical issues were also reported. These related to VR headsets: *"Making sure the batteries didn't die on the VR because they were used back-to-back"* (R1), poor graphics due to ill-fitting headset: *"Even people with glasses. I think we had a situation where you know the screen went blurry for them"* (R1), and researchers having to use their personal Facebook™ accounts to sign in: *"The Oculus™ [VR headset brand] portal. . .that is kind of a bit of a tricky situation. . .when you can't make dummy Facebook™ accounts for it, you have to use personal Facebook™ accounts"* (R2). Another challenge relating to E-MAT$_{VR}$ was the risk of nausea: *"I think, maybe one or two people on the day. . .maybe one guy in particular got very nauseated from it"* (R3).

*Resources incurred.* Researchers discussed the obstacles they encountered while conducting data collection, specifically in relation to lack of time and personnel available: *"Trying to do fidelity checks at the same time while they were doing the interventions was actually a little bit tricky"* (R1). Therefore, *"allocating the time"* (R1) for participants, ensuring there was *"enough research members there"* (R1), and having *"more manpower. . .somebody doing the checks specifically"* (R2) were highlighted as key during data collection. Another challenge related to recruiting participants, particularly among those who initially registered their interest in participating but did not respond to the invite to participate: *"When they [participants] are confronted with the sort of the email to say 'oh thank you for signing can we organise something', that initial interest maybe drops off"* (R2).

*Surveys at T0 and T1*. Researchers felt that some survey questions at T0 and T1 were *"similar"* (R3, R4) and *"overlapping"* (R3). This led to confusion among some participants: *"A few people did get a bit confused with the T1 questions. . .they thought they were answering the same thing again"* (R4). Therefore, explaining briefly to participants that the questions are similar in the surveys was an important lesson learned: *"Just mention that like these are going to be similar questions again you already answered. . .it takes a second to say it to them"* (R4). In addition, the surveys did not capture *"technical feedback"* (R4) or information about the *"usability of the software"* (R4). However, one researcher noted that *"putting extra* [questions] *would have been impossible"* (R4) due to the length of the surveys at T0 and T1.

*Recommendations for future research*. Researchers recommended incorporating VR into education and learning by *"bringing them* [VR headsets] *around to secondary schools. . .GAA clubs"* (R4), and having a designated location *"where people can learn in their own time"* (R4) such as *"*[University] *library. . .so if someone wants to spend time they can book that out and they have the headsets"* (R4). Making VR accessible was also perceived as important, with one researcher discussing the need to *"bring them* [VR headsets] *to the people. . .people aren't very likely to download them"* (R4) so *"it* [E-MAT$_{VR}$] *has to exist somewhere where people can go to it and. . .freely"* (R4). Ensuring VR is accessible to everyone and not just GAA players was also discussed: *"I would go for sports players in general. . .not just limited to GAA"* (R4). Accommodating different levels of VR experiences among participants was also advised by researchers, particularly for *"those who struggled with the VR"* (R1) and needed *"a longer tutorial"* (R1) or an *"extra bit of guidance of at the start"* (R2). Going forward, one researcher recommended having *"a process set in stone. . .to say that this is how we do it. . .and this is how we deal with people who aren't so used to VR"* (R4). Another key recommendation by researchers was to have *"more training"* (R1, R4) for the use of Castor EDC software for data collection to help make the process *"a little more. . .streamlined"* (R2). Finally, one researcher suggested more *"user testing"* (R4) to help *"find bugs"* (R4) in E-MAT$_{VR}$. Another researcher suggested *"tweaking"* (R3) the survey questions to get more *"concise"* (R3) *and* *"insightful"* (R3) answers.

## Pilot efficacy outcomes

Across all participants, mean knowledge scores increased from 0.4 (SD 0.2) at baseline to 0.8 (SD 0.2) at T1. At T2, overall mean scores for participants were 0.7 (SD 0.2). However, the mean knowledge scores for E-MAT$_{VR}$ versus E-MAT$_E$ participants did not differ at any timepoint (Table 5).

At baseline, 54.1% (n = 20) in E-MAT$_{VR}$ versus 24.3% (n = 9) in E-MAT$_E$ purposefully felt or examined their testes within the past month (OR 3.7, 95%CI 1.4 to 10.2, p = 0.01) (S5 Table in S3 File). At T2, all E-MAT$_{VR}$ participants and most E-MAT$_E$ participants (n = 29, 90.6%)

**Table 5. Distribution of knowledge scores by trial arm and time point.**

| Characteristic | N* | Overall N = 74** | Study Arm | | Difference in means (95% CI)*** | p-value |
|---|---|---|---|---|---|---|
| | | | E-MAT$_E$ N = 37** | E-MAT$_{VR}$ N = 37** | | |
| Knowledge score at T0 | 74 | 0.4 (0.2) | 0.4 (0.2) | 0.4 (0.2) | 0 (-0.1 to 0.1) | 1 |
| Knowledge score at T1 | 74 | 0.8 (0.16) | 0.8 (0.14) | 0.8 (0.18) | 0 (-0.1 to 0.1) | 1 |
| Knowledge score at T2 | 66 | 0.7 (0.2) | 0.7 (0.2) | 0.7 (0.2) | 0 (-0.1 to 0.1) | 0.94 |

* N = 74 participants at T0 and T1 and 66 at T2 due to 8 participants lost to follow up.

** Mean (SD)

*** Treatment effect estimated using linear regression. Models for knowledge scores at T1 and T2 are adjusted for baseline knowledge scores.

purposefully felt or examined their testes within the past three months. Additionally, in the past three months, 61.7% (n = 21) of E-MAT$_{VR}$ participants versus 71.9% (n = 23) of E-MAT$_E$ participants advised at least one man about the importance of feeling and examining their testes in the shower or bath at least once (OR 0.6, 95%CI 0.2 to 1.8, p = 0.39) (S6 Table in S3 File).

The internal consistency of multi-time scales was generally high and in line with what has been previously reported (S7 Table in S3 File) [24]. Additional information on all participant outcomes can be found in S8-S11 Tables in S3 File.

## Discussion

Fewer than 50% of trials meet their recruitment targets [52]. The top reasons for recruitment failure and high attrition rates include overoptimistic recruitment targets, narrow eligibility criteria, lack of researcher engagement, lack of researcher training and experience in recruitment, insufficient funding, and high participant burden [53]. This highlights the importance of conducting feasibility studies inclusive of process evaluations, hence our study.

We set our sample size at 59 [45]. Bearing in mind a potential attrition rate of 25%, we recruited 74 participants within six months. Only 10.8% of participants were lost to follow up, yielding a final sample size of 66 participants. Successful recruitment and the low attrition rate can be attributed to several factors. These include but are not limited to: Having a dedicated budget for social media advertising/recruitment; involvement of two patient and public contributors in recruiting participants from their respective GAA clubs; developing a standard operating procedure to ensure the standardisation of data collection among the four researchers across all data collection sites; the trial being categorised as low risk; the use of innovative technology namely VR that is appealing to younger men; collecting data in GAA clubs taking into account participants' availability/preferences; contacting participants at T2 using their preferred contact method(s); sending automated reminders to participants to complete the third and final survey at T2 (i.e., an initial e-mail with two reminders); and giving participants who complete the final survey a gift voucher.

As for the instruments, completion was acceptable with little missing data. Primary and secondary efficacy outcomes improved regardless of study arm. This is encouraging, particularly in the context of educating hard-to-reach populations such as young and healthy men. Given that the aim of a feasibility study is to examine the feasibility of conducting a definitive RCT, estimated effects based on between-arm contrasts in outcomes are exploratory and aimed at informing the design of a future RCT [26]. That said, with respect to knowledge scores, the observed data showed no benefit of E-MAT$_{VR}$ over E-MAT$_E$. This is not unexpected in a small feasibility study.

The latest MRC framework for developing and evaluating complex interventions calls for researchers to consider six core elements relating to the process of implementation [25]. These include how the intervention interacts with the context; how it is underpinned by programme theory; what are the diverse stakeholder perspectives; what are the key uncertainties; what is the potential for the intervention to be refined; and are there comparative resource and outcome consequences of the intervention [25]. Findings from the embedded mixed method process evaluation indicate that participants were favourable towards the feasibility and application of the intervention/control. Furthermore, high levels of satisfaction with the quality, delivery, and experience were noted. Findings from the current study provide a framework of learning towards the effective implementation of a larger RCT.

Implementation science frameworks and logic models enunciate the importance of understanding the influence of context and external influencing factors in establishing the feasibility of implementing an intervention [54]. These factors support implementation by mitigating

certain factors of intervention failure. In the current study, recruitment was assisted by the availability and promotion of a small bursary, enthusiastic champions/patient and public collaborators, access through GAA clubs, and the novelty of the intervention. These inputs and activities resulted in high levels of engagement, follow-up, and satisfaction with the intervention. This complements Nilsen and Bernhardsson's [54] scoping review where authors found that the most extensively cited context dimensions included "organizational support, financial resources, social relations and support, and leadership" (p. 1). Current findings highlighted the need for a tailored recruitment strategy with language that is appealing to potential participants and getting 'buy in' and agreement with partner organisations in advance. Other participants noted that VR is helpful across education and in any context.

The current study contributes to advancing knowledge of online and VR interventions in a variety of settings [55–57]. The fact that many participants found the intervention to be a *"fun"* way to learn, *"engaging"* and *"interactive"* with some seeking more interactivity fits with the Preconscious Awareness to Action Framework underpinning the E-MAT$_{VR}$ intervention [38, 39] and the research team's efforts to create explicit memories which combine episodic memories (i.e., memorable events and activities such as VR gaming) with semantic memories that is general knowledge on health promotion. Ultimately, it is the research team's aspiration that men are facilitated to increase their conscious and unconscious awareness of testicular diseases through VR.

There is universal acceptance that health promotion is an effective tool for global health; however, there is less certainty on the tools and mechanisms of achieving this [58]. VR offers great promise. However, the practicality of using VR as a health promoting tool for large numbers of participants was highlighted as a challenge. This could also be seen as a positive as VR offers the potential to ensure fairer access to and from hard-to-reach people and places [59]. In addition, the constant evolution of technology creates additional challenges for researchers and clinicians in keeping interventions up to date [59].

Two key recommendations relating to E-MAT$_{VR}$ refinement were cited by study participants. Participants recommended enhancing the interactivity and visual content of the intervention. This desire for increased interactivity has been observed across other feasibility tests of web-based or digital interventions [60]. Voorheis et al. also highlighted the importance of incorporating the Behavioral Design and Design Thinking theory into mobile health interventions enunciating the importance of incorporating user-centred features and behaviour change content [61]. In addition, participants noted that the reach of the intervention may be improved by targeting male-specific environments. This complements men's mental health promotion interventions (n = 25 papers) which highlighted the importance of both work-based interventions and gender-sensitive interventions which together have the potential power of giving permission to other men "in norming conversations about ordinarily private health matters" (p. 1834) [62].

Finally, the perceived effectiveness of intervention components, the feasibility of the study and intervention procedures, the relations between implementation, mechanisms, and context, and the barriers and facilitators were all examined in this evaluation. In terms of feasibility, participants felt that the intervention was suited to men from different ethnic backgrounds, although this was not explicitly tested, and participants were ethnically homogenous. Furthermore, participants were satisfied with the time taken to complete the intervention and the proportion of participants who participated as a percentage of those who registered interest to participate in the intervention was high (84%). Participants extolled the potential utility of VR as an educational tool in many contexts including schools and hospitals whilst also acknowledging that there may be limited contexts in which this type of intervention is applicable. Although a relatively new concept which requires further

research [55], VR interventions may be an important means of meeting demand to increase health promoting activities when human resources are over-stretched. Furthermore, they provide an alternative opportunity in delivering health promoting information to hard-to-reach populations [63, 64].

## Strengths, limitations, and lessons learned

The pilot feasibility study and mixed method approach to the process evaluation enhanced the data and enabled wider perspectives on the VR intervention to be appreciated. This was a pilot feasibility study with an embedded process evaluation which signposted the way for a future RCT. Modifications for both, intervention delivery and research processes were highlighted which will make a future RCT more likely to achieve its target recruitment and provide valuable research evidence that will facilitate actionable future recommendations for testicular self-examination.

Our study, however, is not without limitations. Below is a list of lessons learned to address the limitations. While this list is not exhaustive, it is valuable for the scaling up of the interventions and the conduct of a future RCT. Lessons learned are also of relevance to researchers who are planning on testing the feasibility of complex technology-based health promoting interventions.

**Lesson 1: Using virtual reality to promote the health of young men.** The number of VR users is growing. The global VR market size is projected to increase from <12 billion U.S. dollars in 2022 to >22 billion U.S. dollars by 2025 [65]. Advancements in VR have led to the development of user-friendly, lightweight, affordable, accessible, and wireless VR technology [42]. Most VR users (94%) are aged 18 to 54 years, with the majority (60%) being men [66]. Therefore, VR can serve as a tool to engage hard to reach populations (i.e., younger men) in health promotion initiatives. This could explain the high satisfaction and the high retention rate (89.2%) among our participants. Having said that, while testicular knowledge and testicular self-examination behaviour improved significantly among all participants, these outcomes did not differ significantly by trial arm. However, this was a feasibility trial and a full RCT, with a sample size powered to detect the efficacy of the intervention would be needed to establish efficacy.

**Lesson 2: Estimating resources incurred by the interventions.** Interactive digital interventions tend to outperform non-interactive ones [67]. Such interventions, however, are expensive. Indeed, resources incurred in our study were identified by researchers as a potential challenge for a future RCT. Therefore, investigating the cost effectiveness of E-MAT$_{VR}$ versus E-MAT$_E$ is important to inform policy, resource allocation decisions, and the design and conduct of a future RCT. This would involve, for example, estimating intervention delivery costs inclusive of personnel and material, investigating healthcare utilisation regarding testicular disease at the conclusion of the RCT, and estimating participants' willingness to pay for E-MAT$_{VR}$ and E-MAT$_E$. It is notable that VR as a medium is becoming much more mainstream with the target population more likely to own VR devices in the future which will help with costs [42, 65, 66].

Of note, a feasibility economic evaluation was conducted as part of our study to explore the practicality of conducting a full-scale economic evaluation in a future RCT. Results from the feasibility economic evaluation have been analysed but not published yet. Initial results, however, suggest that we proceed with an economic evaluation for a full RCT.

**Lesson 3: Revisiting and refining the intervention.** Some minor technical issues, with a very small number of participants, may have impacted their experience of the intervention. Therefore, there is value from revisiting E-MAT$_{VR}$ to help identify technical issues/glitches.

There is also value from incorporating more visual content, interactive elements, and gamification features to maximise engagement and potentially improve the effectiveness of E-MAT$_{VR}$.

**Lesson 4: Accounting for confounders.** Given that the focus of our study was on process and feasibility rather than efficacy, we decided from the outset not to exclude participants who had a personal or family history of testicular disease. We do acknowledge, however, that such individuals may be more familiar with signs, symptoms, treatment, burden, and the overall experience of a patient with testicular disease. Furthermore, participants were GAA players and coaches who are more susceptible to testicular injuries and consequent disease. These individuals could also be more aware of testicular diseases. However, one could argue that given their susceptibility to the disease, the target population is appropriate. Nevertheless, these factors could have affected participants' knowledge at baseline. Notwithstanding this potential confounder, only 2 of the 74 participants reported a personal history of testicular disease. The inclusion of participants who have a personal and/or family history of testicular disease ought to be considered in the design of a future RCT.

**Lesson 5: Ensuring diversity within the sample.** The limited ethnic diversity within the sample may have limited our ability to assess the potential problems with the experience of the research process or the intervention in a more diverse sample. This arises from the relatively ethnically homogenous population in Ireland, though it is changing, and is particularly ethnically homogenous for indigenous sports like the GAA. Therefore, there is value from expanding the study population beyond the GAA, to include a more diverse sample of men as well as targeting more heterogenous male-dominated environments.

**Lesson 6: Encouraging patient and public involvement.** The importance of patient and public involvement in health research is widely recognised and advocated [68]. The contribution of our public and patient representatives (AO'C, MO'R) was instrumental in ensuring that study documents were clear, and the study was relevant to younger men. In addition, both public and patient representatives played a key role in recruiting several participants from their GAA clubs. This strength ought to be leveraged while planning recruitment in a future RCT.

**Lesson 7: Incentivising participants.** As identified by participants and researchers, the provision of a conditional incentive at the conclusion of the study was one of the key contributors to the high retention rate. This warrants consideration in a future RCT. Of note, the model of payment that we used is an appreciation model offered to participants at the end of the study, therefore avoiding undue inducement [69].

**Lesson 8: Examining survey length and content.** Lengthy and repetitive surveys tend to lead to missing data, participant disengagement, and drop-out [70]. Indeed, information overload and repetition within the surveys were identified as problematic by some of our participants and researchers, despite having a panel of patient and public representatives review the instruments *a priori*. To this end, there is value from carefully examining the survey length and content. This would help ensure optimal participant engagement and data collection in a future RCT.

**Lesson 9: Using qualitative research.** Qualitative research alongside RCTs provides added value [71, 72]. In our study, qualitative research, particularly in the process evaluation, helped provide an in-depth account of men's experiences of participating as well as researchers' experiences of delivering the interventions. Qualitative research also informed the lessons learnt which will be considered in the design and conduct of a future RCT.

## Conclusion

Our study highlights the feasibility, context, fidelity, usability, and satisfaction associated with the implementation of a VR intervention in relation to testicular diseases among young

athletes. The intervention has the potential to engage this audience in health education [23]. Findings from our study are promising. Considering the lessons learned, some modifications are required to the E-MAT interventions prior to conducting a future definitive RCT comparing the effect of E-MAT$_{VR}$ to E-MAT$_E$. Overall, this study has provided data which supports the potential utility of VR as an education and health promotion tool in a healthcare context for a hard-to-reach population.

## Supporting information

**S1 File. Study protocol.**
(DOCX)

**S2 File. CONSORT 2010 checklist of information to include when reporting a pilot or feasibility trial.**
(DOCX)

**S3 File. S1-S11 Tables.**
(DOCX)

## Acknowledgments

We would like to thank the Chairpersons, Healthy Club Officers, and Public Relations Officers in the participating Gaelic Athletic Association (GAA) clubs for helping with recruitment. We would like to acknowledge Mr. Cathal O'Sullivan who helped with data collection. We would also like to thank the Health Research Board Clinical Research Facility, University College Cork for data management support and Mr. Patrick Hayes for designing the study logo and recruitment poster. Finally, we would like to acknowledge the Trial Steering Committee Members Dr Aine O'Donovan and Dr Jack Wilkinson who helped ensure that the trial was being conducted safely and to the highest standard.

## Author Contributions

**Conceptualization:** Mohamad M. Saab, Martin P. Davoren, Frances Shiely, Janas M. Harrington, Gillian W. Shorter, David Murphy, Eoghan Cooke, Aileen Murphy, Michael J. Rovito, Steve Robertson, Serena FitzGerald, Alan O'Connor, Josephine Hegarty, Darren Dahly.

**Data curation:** Megan McCarthy, Darren Dahly.

**Formal analysis:** Mohamad M. Saab, Megan McCarthy, Martin P. Davoren, Billy O'Mahony, Eoghan Cooke, Josephine Hegarty, Darren Dahly.

**Funding acquisition:** Mohamad M. Saab.

**Investigation:** Mohamad M. Saab, Megan McCarthy, Billy O'Mahony, Eoghan Cooke, Ann Kirby, Alan O'Connor, Mícheál O'Riordan.

**Methodology:** Mohamad M. Saab, Martin P. Davoren, Frances Shiely, Janas M. Harrington, Gillian W. Shorter, David Murphy, Aileen Murphy, Michael J. Rovito, Steve Robertson, Serena FitzGerald, Josephine Hegarty, Darren Dahly.

**Project administration:** Mohamad M. Saab, Megan McCarthy.

**Resources:** Mohamad M. Saab, Alan O'Connor, Mícheál O'Riordan.

**Software:** Billy O'Mahony, Eoghan Cooke.

**Validation:** Mohamad M. Saab.

**Visualization:** Mohamad M. Saab, Megan McCarthy, Darren Dahly.

**Writing – original draft:** Mohamad M. Saab, Megan McCarthy, Martin P. Davoren, Josephine Hegarty, Darren Dahly.

**Writing – review & editing:** Frances Shiely, Janas M. Harrington, Gillian W. Shorter, David Murphy, Billy O'Mahony, Eoghan Cooke, Aileen Murphy, Ann Kirby, Michael J. Rovito, Steve Robertson, Serena FitzGerald, Alan O'Connor, Mícheál O'Riordan.

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
