## [Decision Letter · Decision Letter 0]

16 Jun 2024

PONE-D-24-06857Enhancing Men's Awareness of Testicular Diseases (E-MAT) Using Virtual Reality: A Two-Arm Parallel-Group Randomised Pilot Feasibility Study and Embedded Mixed Method Process EvaluationPLOS ONE

Dear Dr. Saab,

Thank you for submitting your manuscript to PLOS ONE. After careful consideration, we feel that it has merit but does not fully meet PLOS ONE’s publication criteria as it currently stands. Therefore, we invite you to submit a revised version of the manuscript that addresses the points raised during the review process.

We look forward to receiving your revised manuscript.

Kind regards,

Alireza Sadeghi, M.D., M.P.H.

Academic Editor

PLOS ONE

Journal Requirements:

"Health Research Board Definitive Interventions and Feasibility Awards (DIFA-2020-028)."

5. We note that you have referenced (Eldridge SM, Lancaster GA, Campbell MJ, Thabane L, Hopewell S, Coleman CL, et al. Defining Feasibility and Pilot Studies in Preparation for Randomised Controlled Trials: Development of a Conceptual Framework. Lazzeri C, editor. PLOS ONE. 2016 Mar 15;11(3):e0150205.) which has currently not yet been accepted for publication. Please remove this from your References and amend this to state in the body of your manuscript: (ie “Bewick et al. [Unpublished]”) as detailed online in our guide for authors

6. We note that Figure 3 in your submission contain copyrighted images. All PLOS content is published under the Creative Commons Attribution License (CC BY 4.0), which means that the manuscript, images, and Supporting Information files will be freely available online, and any third party is permitted to access, download, copy, distribute, and use these materials in any way, even commercially, with proper attribution. For more information, see our copyright guidelines: http://journals.plos.org/plosone/s/licenses-and-copyright.

a. You may seek permission from the original copyright holder of Figure 3 to publish the content specifically under the CC BY 4.0 license. 

Additional Editor Comments:

Dear Authors,

Thank you for submitting your manuscript to us. After careful consideration and consultation with our esteemed reviewers, we have determined that your manuscript would greatly benefit from minor revisions.

In light of this, we kindly request that you revise your manuscript in accordance with the reviewers’ comments and resubmit the updated version through the PLOS ONE editorial manager system.

To facilitate the review process and enhance communication between you and the reviewers, we strongly recommend adding line numbers to your manuscript.

Furthermore, it has been noted by two of our reviewers that there appear to be some discrepancies in your report (comment 10 by Reviewer 5 and comment 14 by Reviewer 1). We kindly ask that you address these issues and provide an explanation for them in your revised submission.

Lastly, we would like to draw your attention to the 7th comment made by Reviewer 5. This particular point requires your special attention during the revision process.

We look forward to receiving your revised manuscript. Thank you for considering PLOS ONE for your publication.

Best regards,

Reviewers' comments:

Reviewer's Responses to Questions

**Comments to the Author**

1. Is the manuscript technically sound, and do the data support the conclusions?

Reviewer #1: Partly

Reviewer #2: Yes

Reviewer #3: Yes

Reviewer #4: Yes

Reviewer #5: Yes

2. Has the statistical analysis been performed appropriately and rigorously? 

Reviewer #1: No

Reviewer #2: Yes

Reviewer #3: Yes

Reviewer #4: Yes

Reviewer #5: Yes

3. Have the authors made all data underlying the findings in their manuscript fully available?

Reviewer #1: Yes

Reviewer #2: Yes

Reviewer #3: Yes

Reviewer #4: Yes

Reviewer #5: Yes

4. Is the manuscript presented in an intelligible fashion and written in standard English?

Reviewer #1: Yes

Reviewer #2: Yes

Reviewer #3: Yes

Reviewer #4: Yes

Reviewer #5: Yes

5. Review Comments to the Author

Reviewer #1: The manuscript is quite well presented but could be further improved.

The title could be improved. For the exclusion criteria, other criteria are to be stated.

The following requires further information.

i) Page 10, the person who collected the info 'T0 and T1 data were collected in person in participants’ respective GAA clubs' is to be stated.

ii) Page 10, the person who performs the randomization.

iii) Blinding information whether just involving the participants and data analyst.

Information on the pilot testing of the researcher-designed instruments prior to their usage is to be described.

The information for the sample size is unclear. More elaboration for the statement ‘detect failures in study processes that occur just 5% of the time’ is to be provided. The primary outcome, effect size, power, significance level, expected proportion of participants experiencing the outcome, sample size prior the consideration of attrition rates is to be stated

For the statement ‘’Qualitative data analysis was conducted by one researcher and checked for accuracy by two researchers to minimise errors and improve data credibility and confirmability’ is the researcher(s)referring to the authors of this study?

Page 19, the analysis of whether intent to treat or per protocol is to be stated.

Information on handling missing (if any) is to be stated.

Table 2, the decimal point for percentage figure is to be standardized e.g. 1 decimal point. Likewise in the text and the supplementary tables.

Page 20, the number of participants who participated in the interview/Focus Group with the researchers for the qualitative study is to be stated.

Page 23, the statement Thirty-six (97%) participants in both E-MATVR and E-MATE either agreed or strongly agreed that the intervention' does not reflect figure(s) in Supplementary Table S2.

For the supplementary tables, ensure the decimal points are standardized and to double-check the p values.

Supplementary Table S2, S4, symbol > for p value to be avoided. Actual p value is to be presented.

Supplementary Table S8, the decimal point for internal consistency is to be reduced.

Table 5, Supplementary Table S9, S12, the discrepancy n=66 and n=74 at T2 is to be denoted in the table footnote.

Ensure all the statistical tests used in the results section are mentioned in the statistical methods in page 19.

For Fisher’s exact test, 1 or 2-tailed test is to be stated.

The accepted level of statistical significance (and 1 or 2-tailed test) is to be stated.

Ensure all information that were reported in the CONSORT checklist are presented/clearly presented in the manuscript.

The numbering for each subtopic/subtitle is to be omitted.

Reviewer #2: 1-I recommend to revise the keywords ; as the "sport" isn't appropriate keyword.

2-The golden time of 6 hours for testicular torsion is not the rule , and this phrase is a little misleading , it depends on degree of torsion , I think it should be replaced with most recent literature

3-I suggest to include all meaningful data in Abstract.

4-I recommend to summarize or merge findings to better presentation of work , it seems the findings reported several times in text.

Reviewer #3: 1. The study design, including the randomized controlled trial (RCT) format, is well-conceived and appropriate for evaluating the feasibility and potential effectiveness of the E-MATVR intervention.

2. The inclusion of a control group (E-MATE) allows for a direct comparison of the two interventions, which is crucial for determining the relative benefits of the virtual reality (VR) approach.

3. The choice of outcome measures, including knowledge scores, self-examination behaviors, and qualitative feedback, is relevant and comprehensive for assessing the feasibility and potential impact of the intervention.

4. The inclusion of both immediate post-test (T1) and 3-month follow-up (T2) assessments provides valuable insights into the sustainability of the intervention's effects.

5. The recruitment strategy, utilizing various digital platforms and physical posters, appears to be effective, as evidenced by the enrollment of 74 participants.

6. The high retention rate of 66 participants (89%) is commendable and suggests that the study procedures and interventions were well-accepted by the target population.

7. The high levels of delivery quality and participant satisfaction reported for both the E-MATVR and E-MATE interventions are encouraging, indicating that the interventions were well-received and feasible to implement.

The feedback on the strengths and challenges of the VR intervention, such as its interactive and engaging nature as well as the potential issues with nausea and technical challenges, provides valuable insights for refining the intervention in the future.

8. The observed improvements in knowledge scores and self-examination behaviors across both intervention groups are promising, suggesting that both approaches may have a positive impact on men's awareness and engagement with testicular health.

9. The maintenance of these improvements at the 3-month follow-up further supports the potential long-term benefits of the interventions.

Overall, the pilot feasibility study and process evaluation provide a solid foundation for the proposed future RCT. The findings are promising and support the continued development and evaluation of the E-MATVR intervention, while also considering the lessons learned from this study to optimize the intervention and study design.

To improve the quality of the article, 1. Explore strategies to further enhance the accessibility and user experience of the VR intervention, addressing the challenges identified in this pilot study.

2. Consider incorporating more interactive elements and gamification features to engage participants and potentially improve the effectiveness of the interventions.

3. Carefully examine the survey length and content to ensure optimal participant engagement and data collection.

4. Expand the study population to include a more diverse sample of men, potentially beyond the current focus on Gaelic game athletes.

I believe this pilot study has successfully demonstrated the feasibility of conducting a future definitive RCT to evaluate the E-MATVR intervention. With the lessons learned and the promising initial results, the proposed RCT is well-justified and has the potential to contribute valuable insights to the field of testicular cancer awareness and prevention.

Reviewer #4: Authors have put together a very nice feasibility study on the use of virtual reality as an educational tool in men's testicular health.

The study is very sound and well written. I have no issues with the premises, nor with the content. There is good recruitment and followup, and I like the addition of a couple of paragraphs at the start of the discussion regarding successful recruitment.

My only comment would be that there does not seem to be any real difference in efficacy between the two arms, suggesting that the use of VR is not necessarily an improvement over more conventional interventions. It would be nice to see this commented on in more detail in the discussion. In particular, the assertion that VR may 'ensure access to hard to reach people and places' and is an 'alternative opportunity in delivering health promoting information to hard to reach populations' would seem counterintuitive, given the additional technology burden and distribution issues with VR. I think these statements would benefit from more consideration and explanation.

Otherwise I have no real issues or corrections. Well done to all authors.

Reviewer #5: Dear esteemed authors,

I would like to express my appreciation for your diligent efforts in producing the article titled "Enhancing Men's Awareness of Testicular Diseases (E-MAT) Using Virtual Reality: A Two-Arm Parallel-Group Randomized Pilot Feasibility Study and Embedded Mixed Method Process Evaluation."The article is well-structured and effectively communicates its findings. Materials and methods are well crafted and meticulously designed. There are no deviations from the registered protocol. The points are articulated and a reasonable flow exists throughout the manuscript. I want to thank you for making your data and code open source, which is a great practice leading to improved reproducibility of the science. However, I have to propose some comments that conceivably improve the quality of your study. Therefore, I believe that a minor revision can benefit the manuscript.

1- Please add line numbering to your manuscript to enhance communications.

2- The sentence "challenges related to getting used to VR, nausea, and technical issues." is a fragment of a sentence and should be edited.

3- A structured abstract would be a better fit for your study. I recommend sectioning your abstract in a way that helps the reader to grasp the essence of your report.

4- The abstract's conclusion is too brief and does not fully communicate the limitations, interpretations, etc. Please trim it and make room for more necessary information.

5- A good proportion of your introduction is devoted to the clinical and epidemiological aspects of the diseases. I think focusing more on details directly related to your topic will greatly enhance the readability and relevance of your study. I suggest to discuss the rationale for the future RCT, the reasons to justify why the pilot trial is needed, and areas of uncertainty that need to be addressed before the future RCT can take place, in depth at the end of your introduction.

6- The ending statement of the introduction's second paragraph:

Your statement comes from a late reference (2014), while currently, there are evidence suggesting watchful observation (aka active surveillance) for testicular cancer. Please update your citation.

7- The study has limited the exclusion criteria to ones with seizure and/or motion sickness. However, in my opinion, a very important characteristic that can affect and potentially, confound the results is having prior encounters with testicular diseases in the person being studied or his close relatives and acquaintances. Such individuals are reasonably and naturally more familiar with clinical manifestations, burdens, treatments and overall experience of a patient with testicular diseases. Furthermore,most of the participants enrolled in the study are players and coaches in the GAA clubs that are more susceptible to testicular injuries and consequent diseases and might be more aware of testicular diseases. The mentioned points might affect not only the main outcomes but also the dose exposure, reach, and internal validity. It may be beneficial to provide justification for not considering such information at recruitment or mention it as a limitation to your study.

8- The study compares an interactive (VR) VS a non-interactive (PDF document) method of education. I assume the first question that comes to the mind of readers (providers and policymakers) is whether VR, which is an expensive technology, can outperform other interactive methods and whether it is cost-effective for them to adopt in in their system. Nonetheless, it is well-known that using interactive models is usually more effective than non-interactive ones. It would be helpful if the authors provide explanations for this point or include it as a limitation of the study.

9- I see the authors provide a lot of supporting information. While this abundance is a good and encouraged practice, I recommend gathering them into a single file with an informative table of contents to facilitate the readers experience. Downloading that many files could be tedious.

10- Table 2 contains redundant information regarding the gender combination of the participants.

11- On page 24, part 3.2.5, the information mentioned in the sentence, "Elements of the VR game that participants did not favor ..." is not the same as those in the supplementary table S5.

12- The discussion could benefit from explaining the limitations in detail and arguing whether and how they could be overcome in the future RCT.

In summary, your article makes a valuable contribution to the field, and I believe the aforementioned suggestions will further enhance it.

6. PLOS authors have the option to publish the peer review history of their article (what does this mean?). If published, this will include your full peer review and any attached files.

Reviewer #1: No

Reviewer #2: No

Reviewer #3: **Yes: **Alireza Jafari

Reviewer #4: No

Reviewer #5: No

---

## [Author Response · Author response to Decision Letter 0]

2 Jul 2024

We would like to thank the Editor and Reviewers of PLOS ONE for their valuable feedback regarding this manuscript. We have uploaded a document titled "Response to Reviewers" where we provided our point-by-point responses to all comments using side-by-side tables. 

We also provide our response below while indicating the page and line numbers in the clean copy of the manuscript. 

JOURNAL REQUIREMENTS

Comment: 1. Please ensure that your manuscript meets PLOS ONE's style requirements, including those for file naming. The PLOS ONE style templates can be found at [two links provided, too long to include here]

Response: We ensured that our manuscript meets all PLOS ONE style requirements. 

Comment: 2. We suggest you thoroughly copyedit your manuscript for language usage, spelling, and grammar. If you do not know anyone who can help you do this, you may wish to consider employing a professional scientific editing service.

Response: This has been done among the authors who are fluent and native English speakers and seasoned academics.

Comment: 3. Thank you for stating the following financial disclosure: 

"Health Research Board Definitive Interventions and Feasibility Awards (DIFA-2020-028)."

Response: The role of the funders was stated in the cover letter of the original submission. We included this statement again in the cover letter for the revised manuscript. 

Comment: 4. Please provide a complete Data Availability Statement in the submission form, ensuring you include all necessary access information or a reason for why you are unable to make your data freely accessible. If your research concerns only data provided within your submission, please write "All data are in the manuscript and/or supporting information files" as your Data Availability Statement.

Response: The data that support the findings of this study are openly available in the Open Science Foundation at https://osf.io/m5wb7/. 

Comment: 5. We note that you have referenced (Eldridge SM, Lancaster GA, Campbell MJ, Thabane L, Hopewell S, Coleman CL, et al. Defining Feasibility and Pilot Studies in Preparation for Randomised Controlled Trials: Development of a Conceptual Framework. Lazzeri C, editor. PLOS ONE. 2016 Mar 15;11(3):e0150205.) which has currently not yet been accepted for publication. Please remove this from your References and amend this to state in the body of your manuscript: (ie “Bewick et al. [Unpublished]”) as detailed online in our guide for authors

Response: We have discussed this with the Senior Staff Editor Miquel Vall-llosera Camps who, and I quote, “discussed this with the Senior Editorial staff and we agree that that reference was published, so our recommendation is to proceed as you suggest and keep that reference. When you resubmit please include in your response to reviewers your explanation for using this reference, as provided here.” 

To this end, the reference to Eldrige et al.’s paper will remain as is. 

Comment: 6. We note that Figure 3 in your submission contain copyrighted images. All PLOS content is published under the Creative Commons Attribution License (CC BY 4.0), which means that the manuscript, images, and Supporting Information files will be freely available online, and any third party is permitted to access, download, copy, distribute, and use these materials in any way, even commercially, with proper attribution. For more information, see our copyright guidelines: http://journals.plos.org/plosone/s/licenses-and-copyright.

Response: We have discussed this with Teresa Diviacchi, Peer Review Operations Specialist. To this end, for Figure 3, I confirm that the screenshots shown are from the E-MATVR intervention and that I hold the copyright for this educational tool. I have submitted the Content Permission Form accordingly.

Comment: 7. Please include captions for your Supporting Information files at the end of your manuscript, and update any in-text citations to match accordingly. Please see our Supporting Information guidelines for more information: http://journals.plos.org/plosone/s/supporting-information.

Response: Captions for Supporting Information files included as recommended and references to these files in the text of the manuscript have been updated accordingly. P44, L1-19. 

Comment: 8. Please review your reference list to ensure that it is complete and correct. If you have cited papers that have been retracted, please include the rationale for doing so in the manuscript text, or remove these references and replace them with relevant current references. Any changes to the reference list should be mentioned in the rebuttal letter that accompanies your revised manuscript. If you need to cite a retracted article, indicate the article’s retracted status in the References list and also include a citation and full reference for the retraction notice.

Response: Reference list revised as recommended. 

EDITOR

Comment: To facilitate the review process and enhance communication between you and the reviewers, we strongly recommend adding line numbers to your manuscript.

Response: Line numbers added as recommended. 

Comment: Furthermore, it has been noted by two of our reviewers that there appear to be some discrepancies in your report (comment 10 by Reviewer 5 and comment 14 by Reviewer 1). We kindly ask that you address these issues and provide an explanation for them in your revised submission.

Response: All reviewers’ comments including comment 10 by Reviewer 5 and comment 14 by Reviewer 1 have been addressed and all discrepancies have been amended/resolved. 

Comment: Lastly, we would like to draw your attention to the 7th comment made by Reviewer 5. This particular point requires your special attention during the revision process.

Response: Thank you for highlighting this comment which we have addressed in full. Please see our response to Reviewer 5 below. 

REVIEWER 1

Comment: The title could be improved 

Response: Shortened as follows: Enhancing Men's Awareness of Testicular Diseases (E-MAT) Using Virtual Reality: A Randomised Pilot Feasibility Study and Mixed Method Process Evaluation. P1, L1-3. 

Comment: For the exclusion criteria, other criteria are to be stated.

Response: In addition to excluding individuals with a history of seizures and motion sickness, we added the following exclusion criteria as suggested: Individuals who were younger than 18 years or older than 50 years, not assigned male at birth, residing outside the Republic of Ireland, and not involved in GAA as either players or coaches were excluded. P10, L8-10. 

Comment: The following requires further information.

i) Page 10, the person who collected the info 'T0 and T1 data were collected in person in participants’ respective GAA clubs' is to be stated.

ii) Page 10, the person who performs the randomization.

iii) Blinding information whether just involving the participants and data analyst.

Response: 

i) Four research personnel collected T0 and T1 data in-person in participants’ respective GAA clubs. P11, L17-18. 

ii) For each participating club, individual participants were randomised with the same probability to one of the two trial arms (E-MATVR or E-MATE) using the Castor EDC software. See new section titled “Randomisation” for the full process. P19, L22-23.

iii) We have provided information regarding blinding in the original submission. However, to ensure clarity and in accordance with CONSORT, we added a new section titled “Randomisation” where we explained randomisation as well as blinding. Given the differences between the two arms, participants were made aware of the arm they were randomised to once they were assigned to either E-MATVR or E-MATE. Outcome assessments were self-reported and returned to research staff with no information on allocation. The data analyst was also blinded to allocation until database lock and completion of the protocol-specified data analysis. P20, L6-10. 

Comment: Information on the pilot testing of the researcher-designed instruments prior to their usage is to be described.

Response: All instruments used in the pilot feasibility trial and most instruments used in the process evaluation were developed, validated, and pilot tested elsewhere [references provided]. P13, L20-24.

As for instruments developed purposefully for this study: interview guides were developed by four senior academics and researchers who have experience in qualitative research (MMS, MMC, MPD, JH). P15, L5-7.

As for fidelity checklists, these were purposefully developed for use in the current study. Checklist items were mapped onto the processes and contents of the E-MATVR intervention and E-MATE control. These checklists were pilot tested with four participants. Data from these participants were not included in the final analysis. No changes were made to these checklists following pilot testing. P16, L26-27 and P17, L1.

Finally, the time it took participants to complete E-MATVR (in minutes) was captured automatically in the VR headset and the time it took participants to complete E-MATE (in minutes) was captured manually by the observing researcher. P17, L2-6.

Comment: The information for the sample size is unclear. More elaboration for the statement ‘detect failures in study processes that occur just 5% of the time’ is to be provided. The primary outcome, effect size, power, significance level, expected proportion of participants experiencing the outcome, sample size prior the consideration of attrition rates is to be stated

Response: The goal of a pilot feasibility study is to identify problems that would impede the conduct of a larger, definitive RCT as opposed to establishing the efficacy of an intervention. There is no gold standard for sample size calculation in pilot feasibility studies and sample sizes as small as 10 and as large as 59 have been recommended (Saab et al., 2018). Julious (2005), for example, suggested a sample size of 12 per arm, whereas Viechtbauer et al. (2015) proposed a formula to calculate the sample size in pilot studies and reported that “if a problem exists with 5% probability in a potential study participant, the problem will almost certainly be identified (with 95% confidence) in a pilot study including 59 participants” (p. 1375). Therefore, we set the sample size at 59 in our study. Allowing for a potential attrition rate of 25%, we aimed to recruit 74 participants. P19, L11-20. 

Comment: For the statement ‘’Qualitative data analysis was conducted by one researcher and checked for accuracy by two researchers to minimise errors and improve data credibility and confirmability’ is the researcher(s)referring to the authors of this study?

Response: Indeed, we clarified that qualitative data analysis was conducted by one author (MMC) and checked for accuracy by two authors (MPD, JH) to minimise errors and improve data credibility and confirmability. P21, L13-15. 

Comment: Page 19, the analysis of whether intent to treat or per protocol is to be stated.

Response: We did estimate between-arm contrasts for the knowledge scores and the testicular self-examination behaviours on an intent-to-treat basis. P20, L14-16. 

Comment: Information on handling missing (if any) is to be stated. 

Response: Missing data were addressed using complete case analysis, following from the small number of missing values. P20, L19-20. 

Comment: Table 2, the decimal point for percentage figure is to be standardized e.g. 1 decimal point. Likewise in the text and the supplementary tables.

Response: All decimal points in the text and tables including supplemental tables are now at 1 decimal point. Table 2 and throughout. 

Comment: Page 20, the number of participants who participated in the interview/Focus Group with the researchers for the qualitative study is to be stated.

Response: All participants were invited to participate in an individual interview with one of the authors (MMC) as part of the process evaluation. Of those, a sub-sample of nine participants agreed to participate (six from E-MATE and three from E-MATVR). Eight interviews were conducted face-to-face and one by telephone. Interviews with participants lasted on average five minutes. Finally, all four researchers who collected data participated in a one-hour virtual focus group that was facilitated by the lead author (MMS). P22, L18-23. 

Comment: Page 23, the statement Thirty-six (97%) participants in both E-MATVR and E-MATE either agreed or strongly agreed that the intervention' does not reflect figure(s) in Supplementary Table S2.

Response: Thank you and apologies for this error. We have now combined Tables S2 and S3 since data in both tables are form the same instrument. We renumbered all supplementary tables accordingly. 

Comment: For the supplementary tables, ensure the decimal points are standardized and to double-check the p values.

Response: All decimal points have now been reduced to 1 decimal point and all p-values have been double checked and reported in full throughout the paper.

Comment: Supplementary Table S2, S4, symbol > for p value to be avoided. Actual p value is to be presented.

Response: Actual p-value stated as recommended. Tables S2 and S3 (formerly S4). 

Comment: Supplementary Table S8, the decimal point for internal consistency is to be reduced. Reduced to 1 decimal point. This is now Table S7.

Response: Reduced to 1 decimal point. This is now Table S7. 

Comment: Table 5, Supplementary Table S9, S12, the discrepancy n=66 and n=74 at T2 is to be denoted in the table footnote.

Response: “N=74 participants at T0 and T1 and 66 at T2 due to 8 participants lost to follow up.” This explanation was added as footnote in Tables 5, S8 (was S9), and S11 (was S12). 

Comment: Ensure all the statistical tests used in the results section are mentioned in the statistical methods in page 19.

Response: Simple between-arm tests for ordinal responses were conducted using Kruskal-Wallis rank sum tests. P20, L23-24.

Comment: For Fisher’s exact test, 1 or 2-tailed test is to be stated.

Response: All reported p-values are based on two-sided tests with statistical significance defined as p < 0.05. 

Simple between-arm tests for ordinal responses were conducted using Kruskal-Wallis rank sum tests. P20, L23-25 and P21, L1-2. 

Comment: The accepted level of statistical significance (and 1 or 2-tailed test) is to be stated.

Response: All reported p-values are based on two-sided tests with statistical significance defined as p<0.05. P21, L1-2. 

Comment: Ensure all information that were reported in the CONSORT checklist are presented/clearly presented in the manuscript.

Response: As recommended, we made sure that all information in the CONSORT checklist is clearly presented in the manuscript. We changed some of the headings/sub-headings accordingly. 

Comment: The numbering for each subtopic/subtitle is to be omitted.

Response: Omitted as recommended. 

REVIEWER 2

Comment: 1-I recommend to revise the keywords ; as the "sport" isn't appropriate keyword.

Response: Thank you for your comment. We deleted the keyword “Sport” and replaced it with “Athletes” to reflect our target population. P5, L12. 

Comment: 2-The golden time of 6 hours for testicular torsion is not the rule , and this phrase is a little misleading , it depends on degree of torsion , I think it should be replaced with most recent literature

Response: Thank you. We replaced this statement with the following: “Testicular torsion requires aggressive management in men presenting with testicular pain that has been ongoing for many hours, even 24 hours or more from the onset of ischemia.” Here, we cited this systematic review: https://pubmed.ncbi.nlm.nih.gov/289

---

## [Editor Report · Decision Letter 1]

5 Jul 2024

Enhancing Men's Awareness of Testicular Diseases (E-MAT) Using Virtual Reality: A Randomised Pilot Feasibility Study and Mixed Method Process Evaluation

PONE-D-24-06857R1

Dear Dr. Saab,

We’re pleased to inform you that your manuscript has been judged scientifically suitable for publication and will be formally accepted for publication once it meets all outstanding technical requirements.

Kind regards,

Alireza Sadeghi, M.D., M.P.H.

Academic Editor

PLOS ONE

Additional Editor Comments (optional):

Dear Respected Authors,

I have meticulously examined your revisions and responses. It brings me great pleasure to inform you that, in my opinion, your article is now suitable for publication. I anticipate that the novel concept presented can make a positive impact on our existing knowledge.

I am satisfied that you have addressed all the comments from the reviewers accurately. However, I noticed that the role of the funder has only been mentioned in the cover letter. I kindly request you to incorporate this information into the main body of the manuscript as it could potentially affect the interpretation of your findings.

Best Wishes,
---

## [Editor Report · Acceptance letter]

11 Jul 2024

PONE-D-24-06857R1 

PLOS ONE

Dear Dr. Saab, 

I'm pleased to inform you that your manuscript has been deemed suitable for publication in PLOS ONE. Congratulations! Your manuscript is now being handed over to our production team.

Kind regards, 

on behalf of

Dr. Alireza Sadeghi 

Academic Editor

PLOS ONE